# Complexin induces a conformational change at the membrane-proximal C-terminal end of the SNARE complex

Ucheor B Choi[1,2,3,4], Minglei Zhao[1,2,3,4], Yunxiang Zhang[1,2,3,4], Ying Lai[1,2,3,4], Axel T Brunger[1,2,3,4]*

[1]Department of Molecular and Cellular Physiology, Howard Hughes Medical Institute, Stanford University, Stanford, United States; [2]Department of Neurology and Neurological Sciences, Howard Hughes Medical Institute, Stanford University, Stanford, United States; [3]Department of Photon Science, Howard Hughes Medical Institute, Stanford University, Stanford, United States; [4]Department of Structural Biology, Howard Hughes Medical Institute, Stanford University, Stanford, United States

**Abstract** Complexin regulates spontaneous and activates $Ca^{2+}$-triggered neurotransmitter release, yet the molecular mechanisms are still unclear. Here we performed single molecule fluorescence resonance energy transfer experiments and uncovered two conformations of complexin-1 bound to the ternary SNARE complex. In the *cis* conformation, complexin-1 induces a conformational change at the membrane-proximal C-terminal end of the ternary SNARE complex that specifically depends on the N-terminal, accessory, and central domains of complexin-1. The complexin-1 induced conformation of the ternary SNARE complex may be related to a conformation that is juxtaposing the synaptic vesicle and plasma membranes. In the *trans* conformation, complexin-1 can simultaneously interact with a ternary SNARE complex via the central domain and a binary SNARE complex consisting of syntaxin-1A and SNAP-25A via the accessory domain. The *cis* conformation may be involved in activation of synchronous neurotransmitter release, whereas both conformations may be involved in regulating spontaneous release.

*For correspondence: brunger@stanford.edu

## Introduction

$Ca^{2+}$-triggered fusion of synaptic vesicles is orchestrated by synaptic proteins, including neuronal SNAREs (soluble N-ethylmaleimide-sensitive factor attachment protein receptor), the $Ca^{2+}$-sensor synaptotagmin, complexin, and other components (*Südhof, 2013*). This work focuses on the cytoplasmic protein complexin and its interactions with SNAREs. Complexin has multiple functions: it activates $Ca^{2+}$-triggered synchronous release, regulates spontaneous release, and increases the primed pool of synaptic vesicles (*Mohrmann et al., 2015*; *Trimbuch and Rosenmund, 2016*). Here we summarize some of the key properties of complexin relevant to this work. Synchronous $Ca^{2+}$-triggered neurotransmitter release depends critically on complexin (*McMahon et al., 1995*), and this activating role of complexin is conserved across all species and different types of $Ca^{2+}$-induced exocytosis studied to date (*Reim et al., 2001*; *Huntwork and Littleton, 2007*; *Xue et al., 2008*; *Maximov et al., 2009*; *Hobson et al., 2011*; *Martin et al., 2011*; *Kaeser-Woo et al., 2012*; *Cao et al., 2013*; *Yang et al., 2013, 2015*). Complexin also regulates spontaneous release, although this effect is less conserved among species and experimental conditions: for example, in *Drosophila* spontaneous release increases with knockout of complexin (*Xue et al., 2009*; *Jorquera et al.,*

**eLife digest** Nerve cells communicate via electrical signals that travel at high speeds. However, these signals cannot pass across the gaps – called synapses – that separate one nerve cell from the next. Instead, signals pass between nerve cells via molecules called neurotransmitters that are released from the membrane of the first cell and recognized by receptors in the membrane of the next. Prior to being released, neurotransmitters are packaged inside bubble-like structures called vesicles. The synaptic vesicles must fuse with the cell membrane in order to release their contents into the synaptic cleft. Proteins called SNAREs work together with other proteins to allow this membrane fusion to occur rapidly after the electrical signal arrives.

Complexin is a synaptic protein that binds tightly to a complex of SNARE proteins to regulate membrane fusion. This protein activates the quick release of neurotransmitters, which is triggered by an increase in calcium ions as the electrical signal reaches the synapse. Complexin also regulates a different type of neurotransmitter release, which is known as "spontaneous release". The complexin protein is made up of different regions, each of which is required for one or more of the protein's activities. However, it is not clear how these regions, or domains, interact with SNAREs and other proteins to enable complexin to perform these roles.

Choi et al. have now investigated whether the different activities of mammalian complexin are related to the structure that it adopts when it interacts with the SNARE complex. Complexes of SNARE proteins were assembled with one of the SNARE proteins tethered to a surface for imaging. Next, a light-based imaging technique called single molecule Förster resonance energy transfer (or FRET) was used to monitor how complexin interacts with the SNARE complex. This technique allows individual proteins that have been labeled with fluorescent markers to be followed under a microscope and can show how they interact in real-time.

Using this approach, Choi et al. showed that complexin could adopt two different shapes or conformations when it binds to the SNARE complex. In one, complexin interacted closely with the SNARE complex so that it made part of the complex change shape. In the other, complexin was able to bridge two SNARE complexes. Complexin can therefore interact with SNARE complexes in different ways by using different regions of the protein.

These findings provide insight into how complexin may regulate membrane fusion via the SNARE complex. In the future, single molecule FRET could be used to study other proteins found at synapses and understand the other steps that regulate the release of neurotransmitters.

*2012*). Likewise, knockdown in cultured cortical neurons increases spontaneous release, although knockout of complexin in mice differently affects spontaneous release depending on the particular neuronal cell type (*Maximov et al., 2009*; *Kaeser-Woo et al., 2012*; *Yang et al., 2013*; *Trimbuch et al., 2014*).

We study here the homolog complexin-1 of the mammalian complexin family whose primary sequence is highly conserved (96%) in mouse, rat, and human. Complexin-1 binds with high affinity (~10 nM) to the neuronal ternary SNARE complex consisting of synaptobrevin-2, syntaxin-1A, and SNAP-25A (*McMahon et al., 1995*; *Chen et al., 2002*; *Pabst et al., 2002*). Complexin consists of four domains (*Figure 1A*): the N-terminal domain (amino acid range 1–27) is required for activation of synchronous $Ca^{2+}$-triggered release, the accessory domain (amino acid range 28–48) is required for regulation of spontaneous release, the central domain (amino acid range 49–70) is required for all functions of complexin-1, and the C-terminal domain (amino acid range 71–134) is involved in vesicle priming and binds to anionic membranes with curvature sensitivity (*Chen et al., 2002*; *Maximov et al., 2009*; *Kaeser-Woo et al., 2012*; *Snead et al., 2014*). The key roles of the four domains of complexin-1 have been reproduced in a reconstituted single vesicle fusion system with neuronal SNAREs, synaptotagmin-1, and complexin-1 (*Lai et al., 2014*). By varying the complexin-1 concentration, this *in vitro* study suggested that the regulatory effect on spontaneous fusion and the activating role on $Ca^{2+}$-triggered release are governed by distinct molecular mechanisms involving different subsets of the four domains of complexin-1. Recent *in vitro* experiments revealed that the accessory domain is entirely dispensable for activation of $Ca^{2+}$-triggered synaptic vesicle fusion,

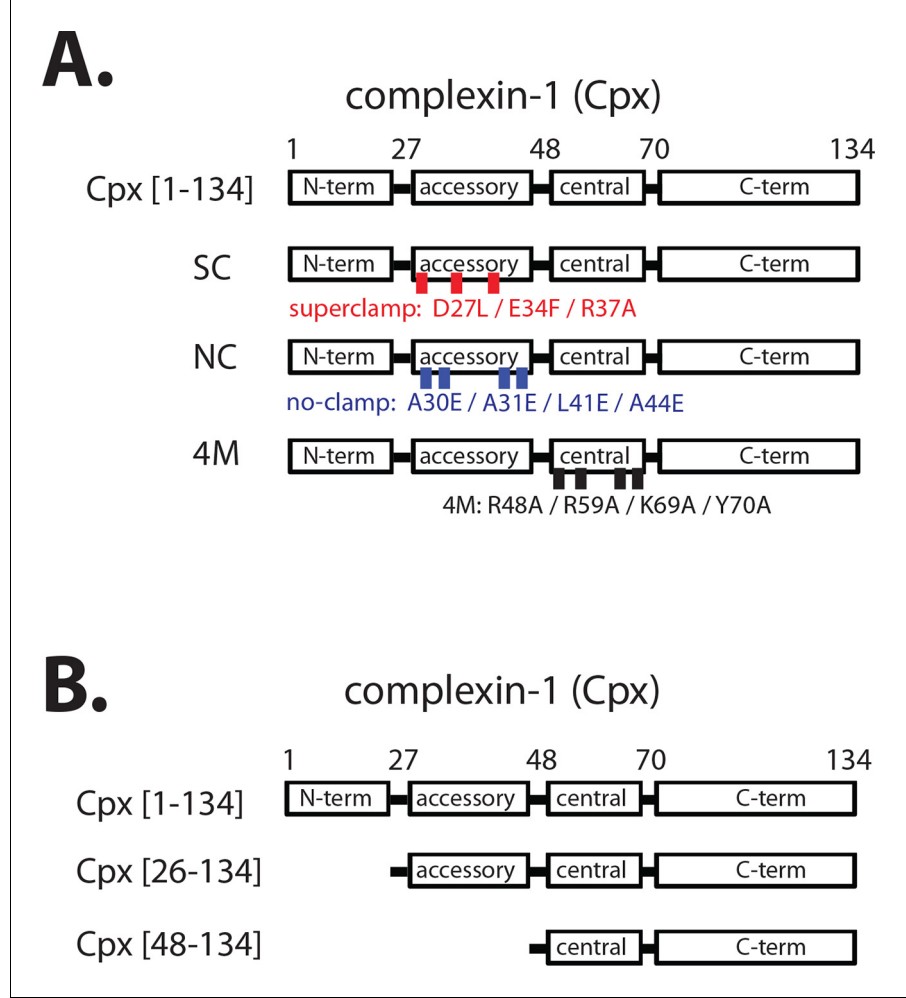

**Figure 1.** Domain diagram of complexin-1, mutants, and truncations. (**A**) Domain diagram for full-length wildtype complexin-1 (Cpx [1–134]) and complexin-1 mutants (SC, superclamp; NC, no-clamp; 4M, mutation of the central helix that prevents binding to ternary SNARE complex). (**B**) Domain diagrams for truncation mutants of complexin-1 (Cpx [26–134], Cpx [48–134]).

although it is essential for regulation of spontaneous release (*Lai et al., 2016*). Similarly, genetic experiments in *Drosophila* also suggest distinct mechanisms for activation of Ca²⁺-triggered release and regulation of spontaneous release (*Cho et al., 2014*).

The central domain of complexin-1 binds to the groove formed by the synaptobrevin-2 and syntaxin-1A α-helices in the complexin-1 / ternary SNARE supercomplex (PDB ID 1KIL) (*Chen et al., 2002*). This interaction is antiparallel, *i.e.*, the direction of the α-helix of the accessory domain of complexin-1 is antiparallel to the direction of the α-helices in the ternary SNARE complex. Functional studies, along with molecular modeling suggested that complexin-1 may inhibit full ternary SNARE complex formation by preventing the C-terminal part of synaptobrevin-2 from binding to the syntaxin-1A / SNAP-25A components of the ternary SNARE complex (*Giraudo et al., 2006*, *2009*). However, a subsequent crystal structure of the 'superclamp' mutant of complexin-1 (D27L, E34F, R37A) (*Figure 1A*) in complex with a partially truncated SNARE complex (containing a truncated synaptobrevin-2 fragment, amino acid range 25–60), along with isothermal titration calorimetry (ITC) and light scattering experiments instead found that complexin-1 bridges two partially truncated SNARE complexes: one partial SNARE complex binds to the central domain of complexin-1 while another partial SNARE complex weakly binds to the accessory domain of the same complexin-1 molecule (PDB IDs 3RK3 and 3RL0) (*Kümmel et al., 2011*; *Krishnakumar et al., 2015*). However, this weak

interaction of the accessory domain was not observable by NMR (*Trimbuch et al., 2014*), and its relevance has been debated (*Trimbuch et al., 2014*; *Krishnakumar et al., 2015*).

Regardless of the ongoing studies of the biophysical interactions involving the accessory domain of complexin-1, this domain is essential for regulation of spontaneous fusion both in neurons and in reconstituted systems. For example, in rescue experiments with complexin knockdown in cultured neurons, the complexin-1 superclamp mutant slightly decreased the mEPSC (miniature excitatory postsynaptic current) frequency, while a 'poor-clamp' mutant (K26E, L41K, E47K) of complexin increased the mEPSC frequency compared to wildtype control (*Yang et al., 2010*). Similarly, genetic rescue experiments in *Drosophila* showed that the superclamp mutant decreased spontaneous release, whereas the no-clamp mutant (L41E, A44E, A30E, A31E) (*Figure 1A*) slightly increased spontaneous release (*Cho et al., 2014*) compared to wildtype control. Moreover, introduction of a helix breaking mutation into the accessory domain of complexin increased spontaneous release in *C. elegans* (*Radoff et al., 2014*).

Different models have been proposed to explain the regulatory function of the accessory domain of complexin on spontaneous release: first, the accessory domain inserts itself into the C-terminal half of a partially assembled SNARE complex and, thus, transiently inhibits full assembly (*Giraudo et al., 2006*, *2009*); second, the accessory domain inserts itself into another SNARE complex, bridging two SNARE complexes (*Kümmel et al., 2011*; *Krishnakumar et al., 2015*); third, the accessory domain is not involved in protein interactions, but instead acts by electrostatic repulsion with the membrane (*Trimbuch et al., 2014*); fourth, the accessory domain stabilizes the α-helix of the central domain of complexin (*Radoff et al., 2014*).

In order to gain more insights into the molecular mechanisms of complexin-1, we studied the complexin-1 / ternary SNARE supercomplex using single molecule fluorescence resonance energy transfer (smFRET) efficiency experiments. We observed two conformations of complexin-1 when bound to the ternary SNARE complex that we refer to as *cis* and *trans*. In the *cis* conformation, the accessory domain cooperates with the N-terminal domain of complexin-1 to induce a conformational change at the membrane-proximal C-terminal end of the bound ternary SNARE complex. In the *trans* conformation, complexin-1 bridges two SNARE complexes: a ternary SNARE complex and a binary SNARE complex, consisting of syntaxin-1A and SNAP-25A (also called t-SNARE complex), requiring the presence of both the central and accessory domains.

## Results

### Proper assembly of SNARE complex

As a prerequisite to study the interaction between complexin-1 and ternary SNARE complex, we wanted to ensure proper assembly of the ternary SNARE complex. SNARE complexes were sequentially assembled on a microscope slide (*Figure 2A* and Materials and methods). The surface of the microscope slide was passivated using biotinylated BSA and a phospholipid bilayer to prevent non-specific binding and to ensure that the tethered proteins are isolated single molecules in an environment surrounded by lipids. Single molecules consisting of the cytoplasmic domain of syntaxin-1A were tethered to the deposited bilayer through a biotin-streptavidin linkage, leading to a more uniform distribution of the tethered proteins on the surface compared to when using reconstituted full-length syntaxin-1A (*Choi et al., 2012*). SNAP-25A and the cytoplasmic domain of synaptobrevin-2 were then added sequentially to assemble the ternary SNARE complex.

In order to assess the proper assembly of ternary SNARE complexes, we designed three FRET label pairs (referred to as SFC1, SFC2, SFC3) with donor and acceptor fluorescent dyes attached to synaptobrevin-2 and syntaxin-1A (*Figure 2B*), respectively. These SNARE label pairs probe the syntaxin-1A / synaptobrevin-2 subconfiguration within the ternary SNARE complex at both ends of the complex. These three label pairs are expected to produce high FRET efficiency for properly assembled ternary SNARE complex based on the crystal structure (PDB ID 1SFC) (*Sutton et al., 1998*). As expected, the smFRET histograms of the three label pairs showed the expected high FRET states, but they also revealed low FRET efficiency states (*Figure 2C*). Nearly 50 percent of the complexes exhibited low FRET efficiency when integrating the corresponding broad peaks (*Figure 2E*). Such low FRET efficiency states had been observed in previous single molecule studies: they are related to improperly assembled SNARE complexes, including antiparallel syntaxin-1A / synaptobrevin-

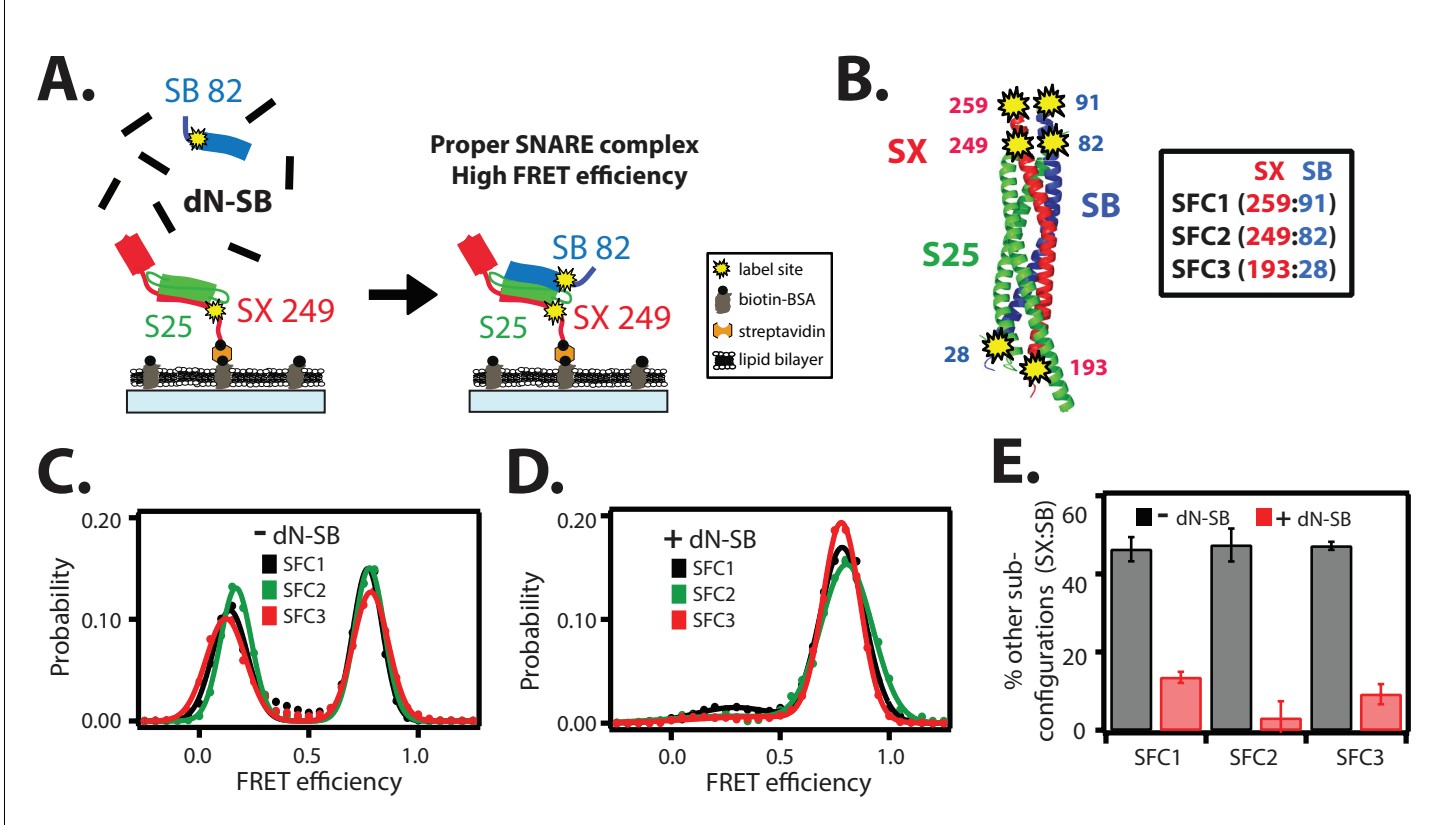

**Figure 2.** The dN-SB method achieves proper assembly of the ternary SNARE complex. (**A**) Left panel: schematic of the dN-SB method for assembly of the ternary SNARE complex. The cytoplasmic domain of syntaxin-1A (SX, colored red) was surface-tethered through biotin-streptavidin (orange dot) linkage to a passivated microscope slide. Next, SNAP-25A (S25, colored green) was added. Subsequently, the cytoplasmic domain of synaptobrevin-2 (SB, colored blue) was added. For the dN-SB method, 10 µM dN-SB fragment was added concurrently with SB to the surface-tethered SX-S25 complex. Unbound proteins were washed away before smFRET measurements. Both SX and SB were labeled with fluorescent dyes (the SNARE label pair SFC2 is indicated by the yellow dots, SX 249 and SB 82). Right panel: properly assembled ternary SNARE complex is expected to produce high FRET efficiency for the SFC2 label pair. (**B**) Location of three SNARE label pairs (SFC1, SFC2, SFC3) in the crystal structure of the ternary SNARE complex (PDB ID: 1SFC), as defined in the legend. Separate experiments were performed for each of the three label pairs. Labeling of the two sites of a particular pair was performed separately with FRET donor and acceptor dyes (Alexa 555 and Alexa 647, respectively, *Figure 2—source data 1*) and the ternary SNARE complex was formed using the dN-SB method. The analysis was restricted to cases where FRET was observed, i.e., complexes that contain one donor and one acceptor dye. (**C,D**) smFRET efficiency histograms for the SNARE label pairs SFC1, SFC2, SFC3 for the surface-tethered ternary SNARE complex that was formed in the absence (**C**) and presence (**D**) of the dN-SB fragment. (**E**) Summary bar chart of the histograms shown in panels **D,E**, illustrating the effect of the dN-SB method in suppressing improper subconfigurations between SX and SB during the assembly of the ternary SNARE complex. "% other subconfigurations (SX:SB)" is calculated as the ratio of the areas under the two Gaussian functions that are fit to the low and high FRET efficiency states in the corresponding smFRET efficiency histograms, respectively. Shown are mean values ± SD for the two subsets of an equal partition of the data that are comprised of the observed FRET efficiency values for all molecules for each different condition (see data summary table in *Figure 2—source data 1*).

The following source data is available for figure 2:

**Source data 1.** Data summary table for the results shown in *Figure 2E*.

2 subconfigurations, when mixed in the absence of other factors that assist proper SNARE complex formation (*Weninger et al., 2003*; *Lou et al., 2015*; *Ryu et al., 2015*).

In order to obtain more uniform ternary SNARE complexes, we added the C-terminal fragment of the cytoplasmic domain of synaptobrevin-2 (referred to as 'dN-SB', amino acid range 49–96 of synaptobrevin-2) during ternary SNARE complex formation (*Figure 2A*). This method had been previously used to obtain more efficient lipid mixing between SNARE-containing liposomes by incubating it prior to *trans* SNARE complex formation (*Pobbati et al., 2006*; *Hernandez et al., 2012*). In earlier work, a similar peptide (V$_c$ peptide, amino acid range 57–92 of synaptobrevin-2) had also been

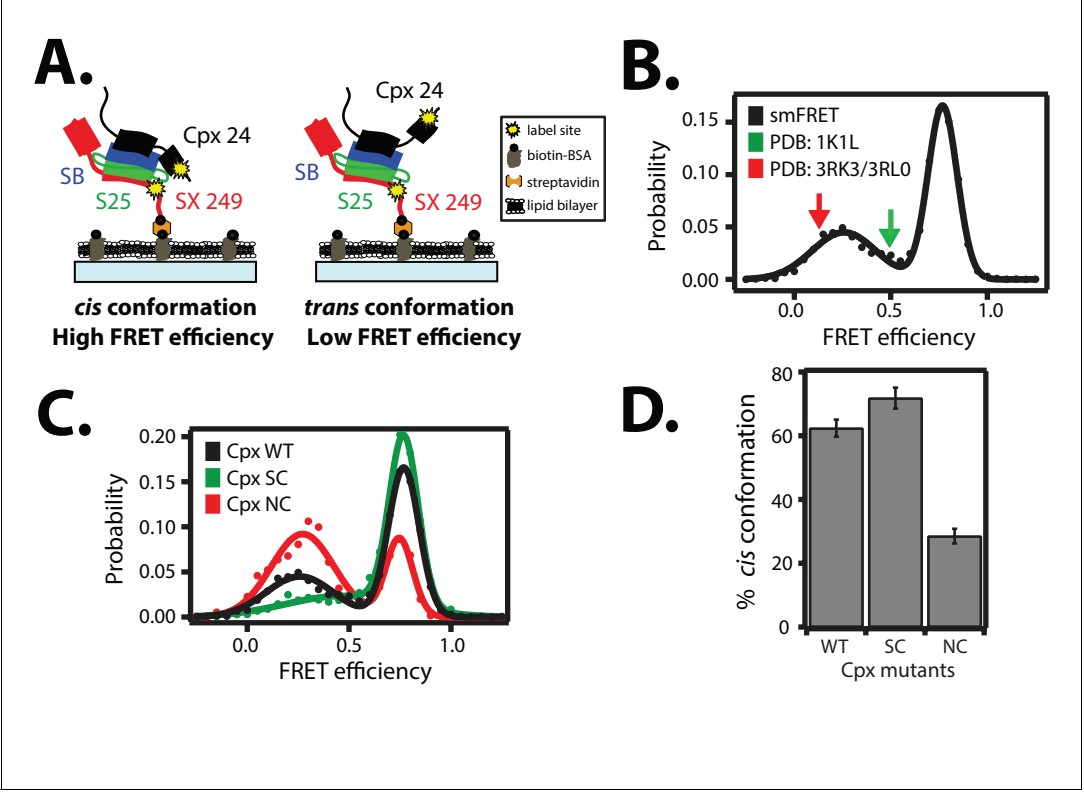

**Figure 3.** *Cis* and *trans* conformations of the accessory domain of complexin-1 when bound to ternary SNARE complex. (**A**) Schematic of smFRET measurements with fluorescent labels attached to the accessory domain of complexin-1 and to the C-terminal end of the ternary SNARE complex. The cytoplasmic domain of syntaxin-1A (SX, colored red) was surface-tethered through biotin-streptavidin (orange dot) linkage to a passivated microscope slide. Prior to tethering SX was labeled with FRET acceptor dye (Alexa 647) at residue 249 (SX 249, yellow dot). Next, SNAP-25A (S25, colored green) was added. Subsequently the cytoplasmic domain of synaptobrevin-2 (SB, colored blue) and 10 µM dN-SB fragment were added concurrently to the surface-tethered SX-S25 complex (i.e., using the dN-SB method). Unbound proteins were then washed away. Full-length wildtype complexin-1 (Cpx, black) was labeled with FRET donor dye (Alexa 555) at residue position 24 (Cpx 24, yellow dot) and added to the surface tethered ternary SNARE complex at 0.01 µM Cpx concentration. Unbound Cpx was then washed away. (**B**) smFRET efficiency histogram for the label pairs described in panel A using wildtype Cpx. Arrows indicate the calculated FRET efficiencies for two crystal structures (red: 0.13 for PDB IDs 3RK3, 3RL0; green: 0.49 for PDB ID 1KIL). (**C**) smFRET efficiency histogram for the label pairs described in panel A using wildtype (WT) Cpx and the accessory domain mutants (SC, superclamp; NC, no-clamp) of Cpx. (**D**) Summary bar chart of the histograms shown in panel C, illustrating the percentage of *cis* conformations for Cpx and its mutants. The "% *cis* conformation" is calculated as the ratio of the areas under the two Gaussian functions that are fit to the high and low FRET efficiency states in the corresponding smFRET efficiency histograms, respectively. Shown are mean values ± SD for the two subsets of an equal partition of the data that are comprised of the observed FRET efficiency values for all molecules for each different condition (see data summary table in *Figure 3—source data 1*).

The following source data is available for figure 3:

**Source data 1.** Data summary table for the results shown in *Figure 3D*.

shown to stimulate lipid mixing (*Melia et al., 2002*). We assembled ternary SNARE complexes in the presence of dN-SB (*Figure 2A*, referred to as 'dN-SB method' in the following). Remarkably, this method nearly eliminated the low FRET efficiency states and the resulting smFRET histograms consisted mainly of one high FRET efficiency state (*Figure 2D,E*).

Our single molecule experiments now provide an explanation of why C-terminal peptides of synaptobrevin produce more SNARE-dependent vesicle fusion when incubating binary (syntaxin-1A / SNAP-25A) SNARE complexes with such peptides prior to *trans* SNARE complex formation (*Melia et al., 2002*; *Pobbati et al., 2006*; *Hernandez et al., 2012*): the presence of the dN-SB fragment during ternary SNARE complex assembly prevents improper syntaxin-1A / synaptobrevin-2 subconfigurations which are probably not fusogenic. We used this dN-SB method for the assembly of ternary SNARE complexes in all subsequent smFRET experiments described in this work.

**Table 1.** γ-corrected smFRET efficiencies between fluorescent dye labels attached to complexin-1 (Cpx) mutants (WT, wildtype; SC, superclamp; NC, no-clamp) and ternary SNARE complex obtained from smFRET measurements. The γ-factors for the Cpx mutants (Cpx WT: 1.66, Cpx SC: 1.62, Cpx NC: 1.42) were empirically estimated, and the corresponding mean values were used to correct the corresponding smFRET efficiency histogram shown in **Figure 3A,B** using **Equation (4)** as described in the Materials and methods. The two FRET efficiencies shown in the table are the peak positions of two Gaussian functions that were fit to the γ-corrected smFRET efficiency histograms, and the error bounds are standard deviations of the peak positions. For comparison, FRET efficiencies were calculated from the specified crystal structures (PDB IDs: 3RK3, 3RL0, and 1KIL, see Materials and methods for the calculation of FRET efficiencies from crystal structures). Cpx was labeled with FRET donor dye (Alexa 555) molecule at residue position 24 and syntaxin-1A was labeled with FRET acceptor dye (Alexa 647) molecule at residue position 249.

|        | γ-corrected smFRET efficiency | FRET efficiency calculated from the specified crystal structures |
|--------|-------------------------------|------------------------------------------------------------------|
| Cpx WT | 0.19 ± 0.12 (*trans*)<br>0.83 ± 0.15 (*cis*) | 0.13 (PDB ID: 3RK3, 3RL0)<br>0.49 (PDB ID: 1KIL) |
| Cpx SC | 0.28 ± 0.18 (*trans*)<br>0.83 ± 0.14 (*cis*) | |
| Cpx NC | 0.24 ± 0.13 (*trans*)<br>0.81 ± 0.12 (*cis*) | |

## *Cis* and *trans* conformations of complexin-1

The two crystal structures of the complexin-1 / ternary SNARE supercomplex (*Chen et al., 2002*; *Kümmel et al., 2011*) suggest that the accessory domain projects away from the SNARE complex at an angle such that it does not interact with the same SNARE complex. Ensemble FRET efficiency experiments also indicated that the accessory domain projects away from the SNARE complex (*Krishnakumar et al., 2011*; *Kümmel et al., 2011*). However, these previous experiments may not have revealed the full conformational space that is available to the complexin-1 accessory domain due to ensemble averaging. In order to fully explore the conformational space of complexin-1 when bound to SNARE complex, we performed smFRET efficiency experiments with fluorescent labels attached to both complexin-1 and the ternary SNARE complex (*Figure 3A*). The ternary SNARE complex was assembled as described in *Figure 2A* using the dN-SB method, starting with the surface-tethered cytoplasmic domain of syntaxin-1A alone, which was labeled with FRET acceptor dye. Unlabeled SNAP-25A and the cytoplasmic domain of synaptobrevin-2 were added sequentially to assemble the ternary SNARE complex. After washing unbound proteins, we added complexin-1 which was labeled with a FRET donor dye within the accessory domain. We observed two populations with low and high FRET efficiency, with their means corresponding to 0.19 ± 0.12 and 0.83 ± 0.15, respectively (*Figure 3B* and *Table 1*). We refer to the high FRET efficiency state as the '*cis*' conformation of complexin-1, whereas we refer to the low FRET efficiency state as the '*trans*' conformation of complexin-1 (*Figure 3A*).

For comparison, we calculated the corresponding FRET efficiencies from the crystal structures of the complexin-1 / ternary SNARE supercomplex ( Materials and methods): the calculated FRET efficiency is 0.49 for the supercomplex between complexin-1 and the fully assembled SNARE complex (PDB ID 1KIL) and 0.13 for the complex between the superclamp mutant of complexin-1 and the partially truncated SNARE complex (PDB IDs 3RK3, 3RL0) (*Chen et al., 2002*; *Kümmel et al., 2011*). Thus, the *cis* conformation of complexin-1 corresponds to a FRET efficiency that is larger than that derived from the crystal structures, including error estimates (*Table 1*). Ensemble averaging in previous ensemble FRET experiments of the complexin-1 / ternary SNARE complex (*Krishnakumar et al., 2011*; *Kümmel et al., 2011*) would have masked the two distinct conformations of complexin-1 that we uncovered: our smFRET data suggests that there are two distinct conformations (*cis* and *trans*) of complexin-1 that both exist when bound to ternary SNARE complex.

We next tested the effect of the so-called superclamp and no-clamp mutations (*Figure 1A*) on the smFRET efficiency histograms (*Figure 3C*). The percentage of the *cis* conformation slightly increased for the superclamp mutant, whereas it substantially decreased for the no-clamp mutant of

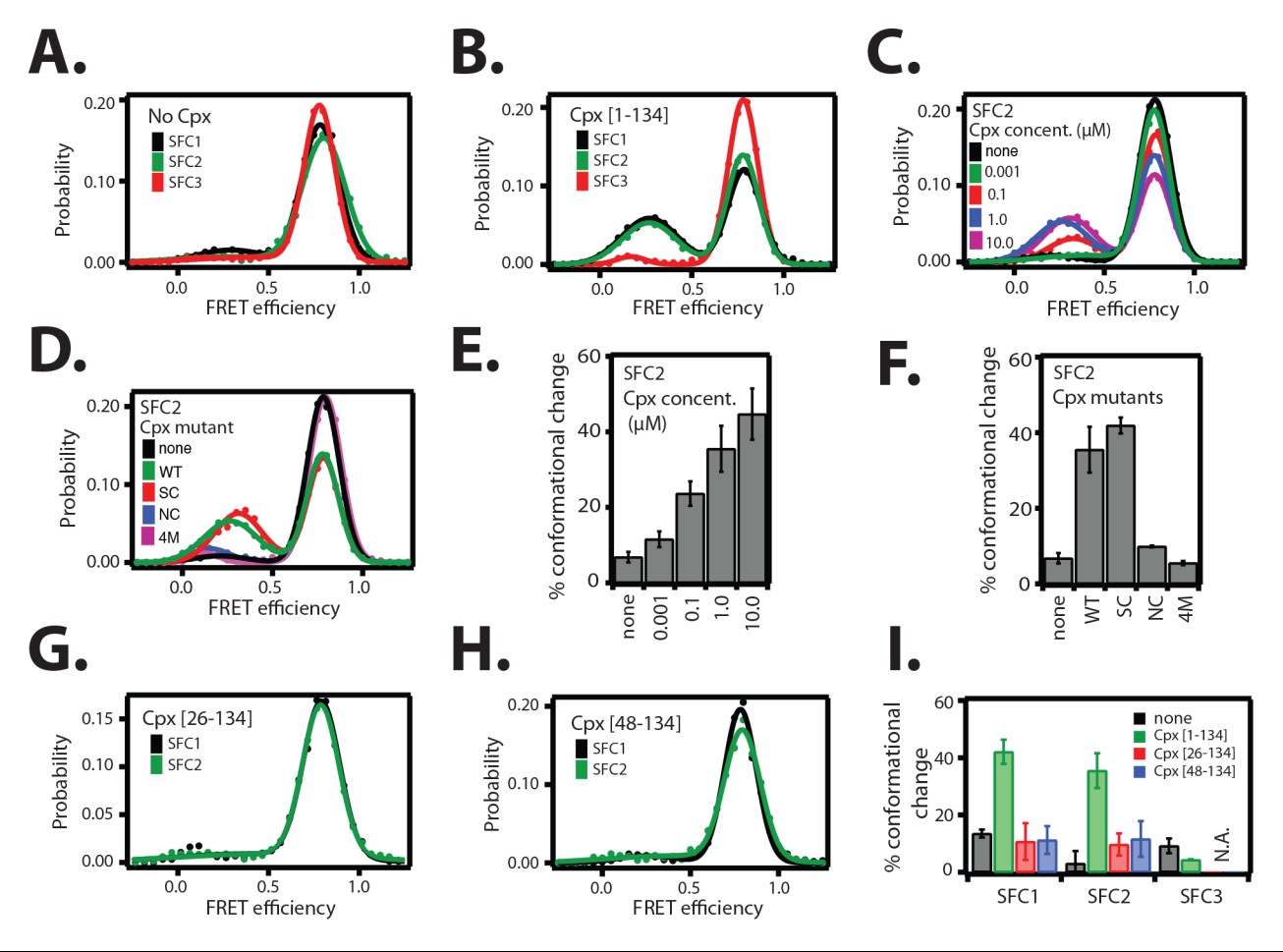

**Figure 4.** The *cis* conformation of complexin-1 induces a conformational change at the membrane-proximal C-terminal end of the ternary SNARE complex. Surface-tethered ternary SNARE complexes were assembled using the dN-SB method as described in *Figure 2A*. The SNARE complex was labeled with each of the three SNARE label pairs SFC1, SFC2, SFC3 (as defined in *Figure 2B*) in separate experiments. Labeling of the two sites of a particular pair was performed separately with FRET donor and acceptor dyes (Alexa 555 and Alexa 647, respectively, *Figure 4—source data 1*) and the ternary SNARE complex was formed using dN-SB method. The analysis was restricted to cases where FRET was observed, i.e., complexes that contain one donor and one acceptor dye. Representative single molecule fluorescence intensity time traces for the SNARE label pair SFC2 are shown in *Figure 4—figure supplement 1*. A data summary table for all experiments in this figure is provided in *Figure 4—source data 1*. (A) smFRET efficiency histograms for SNARE label pairs SFC1, SFC2, SFC3 in the absence of full-length wildtype complexin-1 (Cpx [1–134]) (identical to *Figure 2D*). (B) smFRET efficiency histograms for SNARE label pairs SFC1, SFC2, SFC3 in the presence of full-length wildtype complexin-1 (Cpx [1–134]). 1 μM Cpx [1–134] was then added to form supercomplex with the ternary SNARE complex. (C) smFRET efficiency histograms for the SNARE label pair SFC2 in presence of wildtype full-length complexin-1 (Cpx [1-134]) at the specified concentrations. (D) smFRET efficiency histograms for SNARE label pair SFC2 in the presence of 1 μM full-length wildtype (WT) complexin-1 (Cpx) and its mutants (SC, superclamp; NC, no-clamp; and 4M mutation of the central domain that prevents binding to SNARE complex, *Figure 1A*). (E) Summary bar graph of the histograms shown in panel D. The "% conformational change" is calculated as the ratio of the areas under the two Gaussian functions that are fit to the low and high FRET efficiency states in the corresponding smFRET efficiency histograms, respectively. Shown are mean values ± SD for the two subsets of an equal partition of the data that are comprised of the observed FRET efficiency values for all molecules for each different condition (see data summary table in *Figure 4—source data 1*). (F) Summary bar graph of the histograms shown in panel E. The "% conformational change" is calculated as the ratio of the areas under the two Gaussian functions that are fit to the low and high FRET efficiency states in the corresponding smFRET efficiency histograms, respectively. Shown are mean values ± SD for the two subsets of an equal partition of the data that are comprised of the observed FRET efficiency values for all molecules for each different condition (see data summary table in *Figure 4—source data 1*). (G) smFRET efficiency histograms for SNARE label pairs SFC1 and SFC2 in the presence of 1 μM truncated complexin-1 (Cpx [26–134]) fragment (*Figure 1B*). (H) smFRET efficiency histograms for SNARE label pairs SFC1 and SFC2 in the presence of 1 μM truncated complexin-1 (Cpx [48–134]) fragment (*Figure 1B*). (I) Summary bar chart of the histograms in panels B,C,H, and I, illustrating the effect of WT complexin-1 and its truncation mutants on the conformation of the SNARE complex for the specified label pairs. Label pair SFC3 was not tested for the truncation mutants of complexin-1 since Cpx has no effect on the N-terminal end of the ternary SNARE complex. The "% conformational change" is calculated as the ratio of the areas under the two Gaussian functions that are fit to the low and high FRET efficiency states in the corresponding smFRET efficiency histograms, respectively. Shown are mean values ± SD for the two subsets of an equal partition of the

*Figure 4 continued on next page*

*Figure 4 continued*

data that are comprised of the observed FRET efficiency values for all molecules for each different condition (see data summary table in *Figure 4—source data 1*).

The following source data and figure supplement are available for figure 4:

**Source data 1.** Data summary table for the results shown in *Figure 4A–D,G,H*.

**Figure supplement 1.** Representative single molecule fluorescence intensity time traces for the SNARE label pair SFC2 (Alexa 555 attached to residue 82 of the cytoplasmic domain of synaptobrevin-2 and Alexa 647 attached to residue 249 of the cytoplasmic domain of syntaxin-1A) in the presence of 1 μM full-length wildtype complexin-1 (Cpx [1-134]).

complexin-1 compared to wildtye complexin-1 (*Figure 3D*). The large effect of the no-clamp mutant on the population of the high FRET efficiency state suggests that the *cis* conformation of complexin-1 involves an intimate interaction of the accessory domain of complexin-1 with the ternary SNARE complex.

## The *cis* conformation of complexin-1 induces a conformational change at the membrane-proximal C-terminal end of the SNARE complex

Since the *cis* conformation of complexin-1 suggests an intimate interaction with the membrane-proximal C-terminal end of the ternary SNARE complex, we tested if this interaction affects the conformation of the ternary SNARE complex itself. We used the same SNARE label pairs as described in *Figure 2B* to probe the conformation of the ternary SNARE complex. Upon addition of 1 μM full-length wildtype complexin-1, a low FRET efficiency state emerged for the SFC1 and SFC2 label pairs at the C-terminal end of the ternary SNARE complex, but not for the SFC3 label pair at the N--terminal end (*Figure 4A,B* and *Table 2*) with occasional transitions between high and low FRET states of the SNARE complex (*Figure 4—figure supplement 1*). The population of the low FRET efficiency state increases as the complexin-1 concentration is increased, reaching saturation around 1 μM within experimental error (*Figure 4C,E*). At higher concentration there is a higher likelihood of complexin-1 / ternary SNARE supercomplex formation consistent with the dissociation constant of this complex of around 10–100 nM (*Pabst et al., 2002*; *Li et al., 2007*), and consequently, also a higher likelihood of bound complexin-1 in the *cis* conformation. We note that the low FRET

**Table 2.** γ-corrected smFRET efficiencies between fluorescent labels attached to the ternary SNARE complex (label pairs SFC1 and SFC2 as defined in *Figure 2B*) obtained from smFRET measurements in the presence and absence of 1 μM full-length wildtype complexin-1 (Cpx [1-134]). Mean γ-factors (*Table 2—source data 1*) were used to correct the smFRET efficiency histograms shown in *Figure 4A–B* using *Equation (4)* as described in the Materials and methods. The FRET efficiencies shown in the table are the peak positions of one or two Gaussian functions that were fit to the γ-corrected smFRET efficiency histograms, and the error bounds are standard deviations of the peak positions. As explained in the text, the population of the low FRET efficiency state in the presence of Cpx likely corresponds to the *cis* conformation of bound complexin-1, whereas the population of the high FRET efficiency state likely corresponds to the *trans* conformation of bound complexin-1 or SNARE complex alone. For comparison, FRET efficiencies were calculated from the crystal structure of the ternary SNARE complex (PDB ID: 1SFC, see Materials and methods for the calculation of FRET efficiencies from the crystal structures).

|  | γ-corrected smFRET efficiency - Cpx | γ-corrected smFRET efficiency +Cpx | FRET efficiency calculated from the crystal structure (PDB ID: 1SFC) |
|---|---|---|---|
| SFC1 | 0.85 ± 0.17 | 0.27 ± 0.11<br>0.85 ± 0.14 | 0.98 |
| SFC2 | 0.90 ± 0.18 | 0.31 ± 0.13<br>0.86 ± 0.12 | 0.98 |

**Source data 1.** The means of the empirical γ-factors for SNARE label pairs SFC1 and SFC2 in the presence and absence of 1 μM complexin-1 (Cpx [1-134]) for the data shown in *Figure 5*.

**Table 3.** Fluorescence anisotropy measurements of Alexa 555 or Alexa 647 dyes linked to the specified residues in the cytoplasmic domains of syntaxin-1 (SX) or synaptobrevin-2 (SB), either in isolation or as part of the ternary SNARE complex, in the presence and absence of 1 µM full-length wildtype complexin-1 (Cpx [1-134]). The numbers after 'SX' and 'SB' are the residue positions of the corresponding labeling sites. The formation of ternary SNARE complex and the presence of complexin-1 did not substantially affect the fluorescence anisotropy. Shown are means ± SD for n = 3 replicates.

| Constructs | Steady state anisotropy (r) |
| --- | --- |
| Alexa 555 | 0.20 ± 0.06 |
| Alexa 555 (linked to SB 91) | 0.26 ± 0.01 |
| Alexa 555 (linked to SB 91, in ternary complex) | 0.31 ± 0.01 |
| Alexa 555 (linked to SB 91, in ternary complex) + 1 µM Cpx | 0.32 ± 0.03 |
| Alexa 555 (linked to SB 82) | 0.27 ± 0.01 |
| Alexa 555 (linked to SB 82, in ternary complex) | 0.27 ± 0.02 |
| Alexa 555 (linked to SB 82, in ternary complex) + 1 µM Cpx | 0.28 ± 0.06 |
| Alexa 555 (linked to SB 82, in ternary complex) + 10 µM Cpx | 0.28 ± 0.1 |
| Alexa 555 (linked to SB 28) | 0.24 ± 0.03 |
| Alexa 555 (linked to SB 28, in ternary complex) | 0.32 ± 0.01 |
| Alexa 555 (linked to SB 28, in ternary complex) + 1 µM Cpx | 0.32 ± 0.004 |
| Alexa 647 | 0.15 ± 0.08 |
| Alexa 647 (linked to SX 259) | 0.24 ± 0.09 |
| Alexa 647 (linked to SX 259, in ternary complex) | 0.27 ± 0.1 |
| Alexa 647 (linked to SX 259, in ternary complex) + 1 µM Cpx | 0.27 ± 0.1 |
| Alexa 647 (linked to SX 249) | 0.24 ± 0.02 |
| Alexa 647 (linked to SX 249, in ternary complex) | 0.26 ± 0.1 |
| Alexa 647 (linked to SX 249, in ternary complex) + 1 µM Cpx | 0.27 ± 0.08 |
| Alexa 647 (linked to SX 249, in ternary complex) + 10 µM Cpx | 0.28 ± 0.1 |
| Alexa 647 (linked to SX 193) | 0.27 ± 0.1 |
| Alexa 647 (linked to SX 193, in ternary complex) | 0.26 ± 0.08 |
| Alexa 647 (linked to SX 193, in ternary complex) + 1 µM Cpx | 0.27 ± 0.08 |

efficiency state that is induced by the *cis* conformation of complexin-1 is different from the low FRET efficiency state that is observed when ternary SNAREs are assembled without the dN-SB method (compare *Figures 2C* and *4B*).

To confirm that the change in FRET efficiency is due to a conformational change of the SNARE complex and not caused by dye effects, we followed established procedures in the single molecule field (*McCann et al., 2012*; *Akyuz et al., 2013*; *Munro et al., 2014*) and performed fluorescence anisotropy measurements of the individual fluorophores attached to the SNARE label sites in the presence and absence of full-length wildtype complexin-1 (*Table 3*). There was very little or no change in fluorescence anisotropy for all label sites in the presence of complexin-1 compared to SNARE complex without complexin-1, suggesting that restrictions of the rotational freedoms of the dye molecules by bound complexin-1 are unlikely. Moreover, the sums of the donor and acceptor intensities are similar for the single molecule fluorescence intensity time traces before photobleaching occurs (see representative time traces in *Figure 4—figure supplement 1*), suggesting that there is no large protein induced fluorescence enhancement as seen in some systems when a protein binds near a fluorophore at short distance (*Hwang and Myong, 2014*). We also ruled out photo-physical effects on quantum yield and detection efficiency as an explanation for the observed appearance of the population of the low FRET efficiency state by empirically determining the γ-factor for individual molecules (*Figure 5*, *Table 2—source data 1* and Materials and methods). The low FRET efficiency

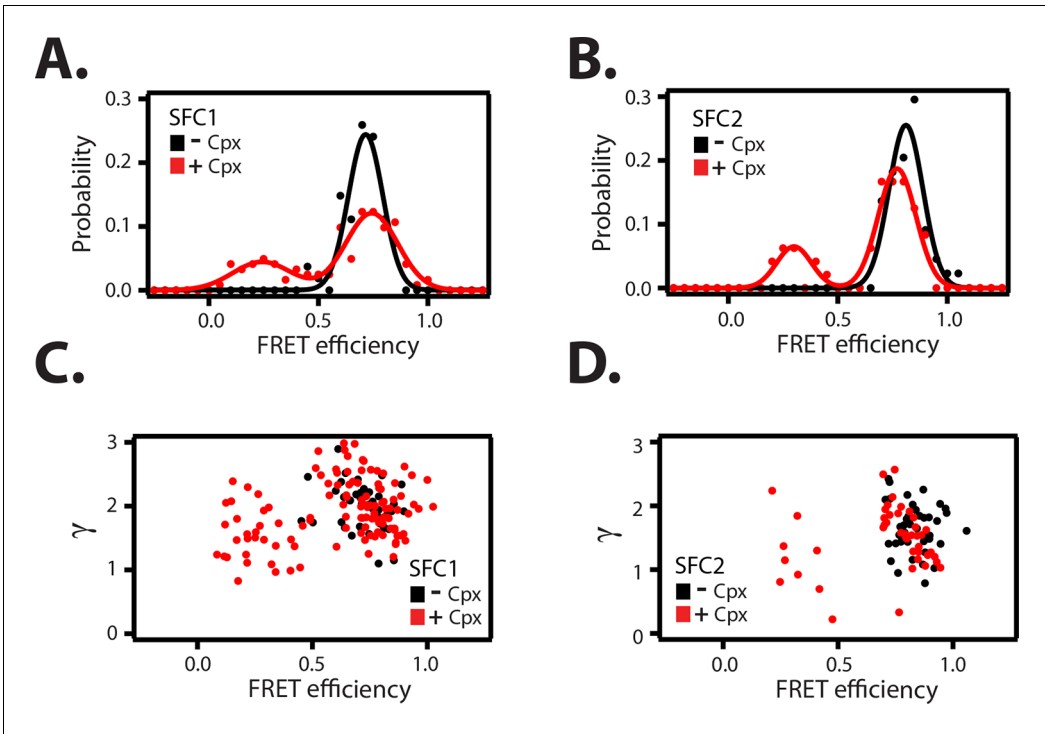

**Figure 5.** Empirical γ-corrected smFRET efficiency histograms. Empirical γ-corrected smFRET efficiency histograms for the SNARE label pairs SFC1 (**A**), SFC2 (**B**), in the presence (red) and absence (black) of 1 µM full-length wildtype complexin-1 (Cpx [1-134]). The FRET efficiency was calculated using *Equation (4)* as described in the Materials and methods. γ *vs.* FRET efficiency for the SFC1 (**C**) and SFC2 SNARE label pairs (**D**), in the presence (red) and absence (black) of 1 µM Cpx [1-134].

state persisted after application of the γ-correction. Finally, we note that measurements of binding kinetics and equilibrium binding constants for the labeled proteins (Materials and methods) agree with literature values of unlabeled proteins (*Pabst et al., 2002*), again suggesting that the fluorophores do not interfere with the binding surfaces. Taken together, all these controls and observations suggest that the population of the low FRET efficiency states observed for the SNARE label pairs SFC1 and SFC2 (*Figure 4B*) represent a distinct conformational state of the ternary SNARE complex that is induced by complexin-1.

We conducted further tests using mutations and truncations of complexin-1. The superclamp mutant (*Figure 1A*) of the accessory domain of complexin-1 slightly increased the population of the low FRET efficiency state compared to wildtype complexin-1, while the no-clamp mutant did not induce a conformational change of the ternary SNARE complex (*Figure 4D,F*). These effects correlate with the FRET efficiency measurements of the complexin-1 / ternary SNARE supercomplex with FRET label pairs attached to complexin-1 and syntaxin-1A (*Figure 3C,D*): there is a slight increase in the population of the high FRET efficiency state by the superclamp mutant corresponding to an increase in the *cis* conformation of complexin-1, whereas the no-clamp mutant decreased the population of the high FRET efficiency state corresponding to a decrease of the *cis* conformation of complexin-1. Therefore the low FRET efficiency state in *Figure 4B* likely corresponds to the *cis* conformation of complexin-1 in *Figure 3B*, and the high FRET efficiency state in *Figure 4B* likely corresponds to the *trans* conformation of complexin-1. Taken together, these experiments support the notion that the *cis* conformation of complexin-1 causes a conformational change at the C-terminal end of the SNARE complex.

As control, we also tested the 4M mutant (*Figure 1A*) of complexin-1 that blocks the binding of the central domain of complexin-1 to SNARE complex and abolishes all functions of complexin-1 (*Maximov et al., 2009*). As expected, it had no effect on the ternary SNARE complex. Finally, the conformational change at the C-terminal end of the ternary SNARE complex

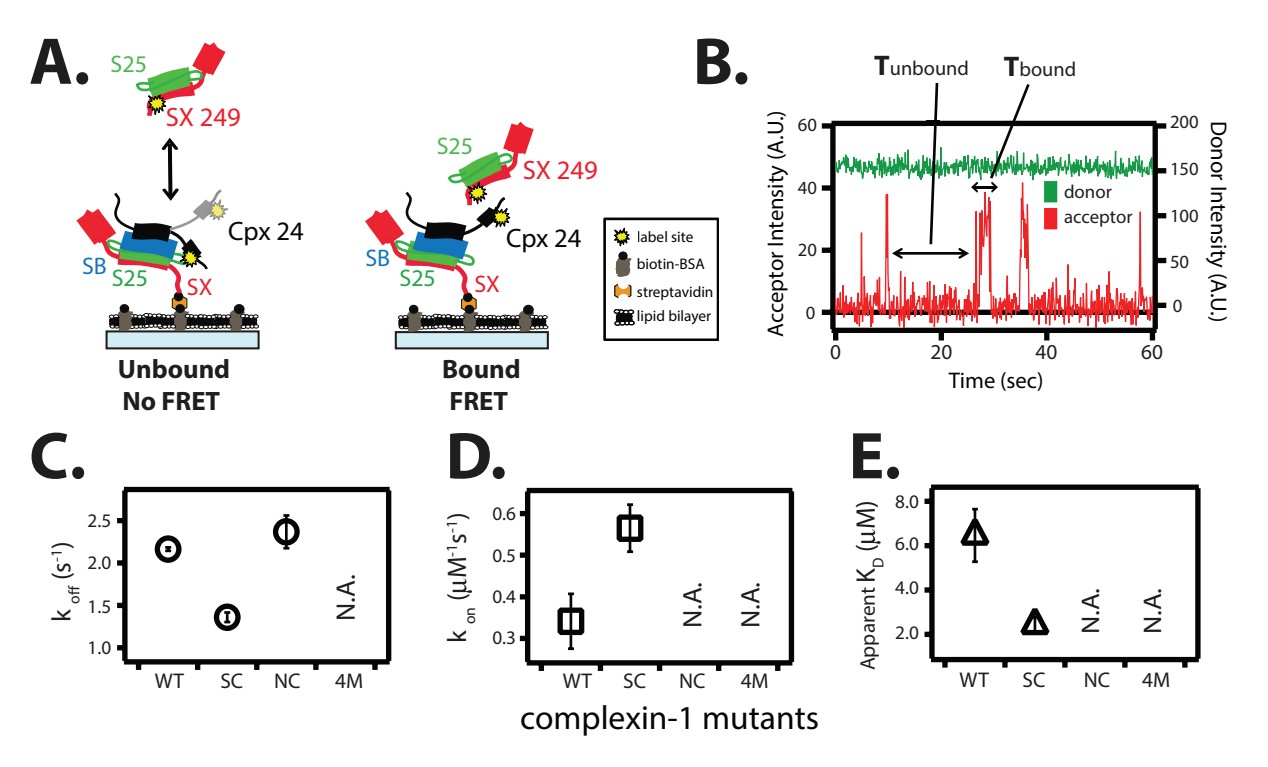

**Figure 6.** The *trans* conformation of complexin-1 can bridge a ternary and a binary SNARE complex through its central and accessory domains. (**A**) Schematic of the single molecule binding experiment. The cytoplasmic domain of syntaxin-1A (SX) was surface-tethered through biotin-streptavidin (orange dot) linkage to a passivated microscope slide. Ternary SNARE complex (consisting of SX, the cytoplasmic domain of synaptobrevin-2 (SB), and SNAP-25A (S25)) was assembled using the dN-SB method. Full-length wildtype complexin-1 (Cpx [1-134]) was labeled with acceptor dye (Alexa 647) at residue position 24 (yellow dot). 50 nM of labeled complexin-1 was then incubated for 5 min to form complex with the surface-tethered ternary SNARE complexes. Unbound Cpx molecules were rinsed away. Next, purified binary SNARE complex (consisting of S25 and SX that was donor dye (Alexa 555) labeled at residue 249 of SX, yellow dot) was added at a concentration of 100 nM to the surface-tethered Cpx / ternary SNARE supercomplex. The appearance of acceptor dye fluorescence intensity indicates FRET from the binary SNARE complex and to the surface-tethered Cpx / ternary SNARE supercomplex. A data summary table for all experiments in this figure is provided in *Figure 6—source data 1*. (**B**) Representative single molecule fluorescence intensity time traces showing smFRET events between the complexin-1 accessory domain of complexin-1 / ternary SNARE supercomplex and a binary (syntaxin-1A / SNAP-25A) SNARE complex. The donor fluorescence intensity is colored green (scale on the right y-axis) and the acceptor fluorescence intensity (scale on the left y-axis) is colored red. Due to the high concentration of the donor labeled proteins in solution, there is no significant effect on the donor intensity upon FRET with an acceptor dye. The stepwise increase in acceptor fluorescence intensity represents bound states and the gaps in between bound states are unbound states. $T_{bound}$ and $T_{unbound}$ represent the dwell time of bound and unbound states, respectively. (**C,D**) Dissociation rates (**C**, open circles) and association rates (**D**, open squares) between binary SNARE complex and surface-tethered Cpx mutants (WT, wildtype; SC, superclamp mutant; NC, no clamp mutant, 4M, mutation of the central helix that prevents binding to ternary SNARE complex, see *Figure 1A*). Rates are calculated as described in Materials and methods. Error bars are standard deviations calculated from two subsets of an equal partition of the data (see data summary table in *Figure 6—source data 1*). (**E**) Apparent dissociation constant $K_D$ (= $k_{off}/k_{on}$) of binding between binary SNARE complex and Cpx or its mutants. Error bars are standard deviations calculated from two subsets of an equal partition of the data (see data summary table in *Figure 6—source data 1*).

The following source data is available for figure 6:

**Source data 1.** Data summary table for the results shown in *Figure 6C–E*.

depended on the inclusion of both the N-terminal and the accessory domains of complexin-1 since truncation of one or both of these domains (*Figure 1B*) did not produce the conformational change (*Figure 4G–I*).

## Complexin-1 in the *trans* conformation can bridge SNARE complexes

The *trans* conformation of complexin-1 observed by smFRET (*Figure 3A*) should be capable of bridging two SNARE complexes as suggested previously (*Kümmel et al., 2011*). To further

corroborate this notion with different SNARE complexes, we first assembled the supercomplex of complexin-1 bound to ternary SNARE complex by tethering unlabeled ternary SNARE complex to the surface of a microscope slide, and then added FRET acceptor dye labeled complexin-1 (*Figure 6A*). After removing unbound complexin-1 molecules, FRET donor dye labeled binary (syntaxin-1A / SNAP-25A) SNARE complex was added in solution and smFRET events were monitored (*Figure 6A*). The observed isolated bursts of acceptor fluorescence intensity reflect binding events of binary SNARE complex to tethered complexin-1 / ternary SNARE supercomplex (*Figure 6B*). From dwell time analysis we obtained rate constants $k_{off} = 2.16 \pm 0.02$ s$^{-1}$ and $k_{on} = 0.34 \pm 0.07$ μM$^{-1}$s$^{-1}$, resulting in $K_D = 6.46 \pm 1.2$ μM (*Figure 6C–E*).

The affinity between the complexin-1 / ternary SNARE supercomplex and the binary (syntaxin-1A / SNAP-25A) SNARE complex is roughly three times weaker than that between complexin-1 and binary (syntaxin-1A / SNAP-25A) SNARE complex alone (*Figure 6E* and Materials and methods). We tested this interaction by mutations of complexin-1. The $k_{off}$ rate of the superclamp mutant of complexin-1 was ~1.5-fold slower ($1.36 \pm 0.06$ s$^{-1}$) than that of wildtype complexin-1, while the $k_{on}$ was ~1.5-fold faster ($0.56 \pm 0.06$ μM$^{-1}$s$^{-1}$), resulting in $K_D = 2.42 \pm 0.3$ μM (*Figure 6C–E*). Thus, the complexin-1 superclamp mutant interacts with the binary SNARE complex with three-fold higher affinity than wildtype complexin-1. In contrast, binary SNARE complex binding events with the no-clamp mutant were very rare within our observation period of 100 s and, consequently, we were unable to determine a dissociation constant for the no-clamp mutant, consistent with the notion that it is the complexin-1 accessory domain that establishes the interaction with the binary SNARE complex in our experiment. As expected, the 4M mutant did not bind at all. In summary, the *trans* conformation of complexin-1 can bridge two SNARE complexes. At variance with this previous study involving partially truncated ternary SNARE complexes (*Kümmel et al., 2011*), our results now show that complexin-1 can also bridge a ternary SNARE complex and a binary SNARE complex.

## Discussion

### Proper assembly of SNARE complexe

Previous single molecule experiments indicated the presence of improperly assembled SNAREs, including antiparallel syntaxin-1A / synaptobrevin-2 subconfigurations, when mixed in the absence of other factors that assist proper SNARE complex formation (*Weninger et al., 2003*; *Lou et al., 2015*; *Ryu et al., 2015*). In some of these studies, the existence of antiparallel SNARE complex subconfigurations were *directly* probed by using antiparallel reporting FRET dye pairs (*Weninger et al., 2003*). SNARE complex was assembled and then purified, followed by reconstitution into liposomes, which were used to form supported lipid bilayers containing reconstituted SNARE complexes. Thus, surface adsorption effects could not have contributed to the observed antiparallel subconfigurations since the ternary SNARE complex was assembled in solution before reconstitution and surface tethering. An independent study (*Ryu et al., 2015*) also suggested the existence of antiparallel SNARE subconfigurations. Using label pairs attached to synaptobrevin-2 and SNAP-25A, a low smFRET efficiency state was observed, consistent with a subpopulation of antiparallel subconfigurations (Figure S4A and supplementary material pages 2–3 for the case without αSNAP in *Ryu et al. 2015*).

As previously described (*Weninger et al., 2003*), antiparallel syntaxin-1A / synaptobrevin-2 subconfigurations can be suppressed by extensive purification of the ternary SNARE complex, including an additional urea wash step. However, such a urea purification step would not be possible with SNAREs that are reconstituted in membranes. Our smFRET experiments indicate that the dN-SB method (*Pobbati et al., 2006*) also results in the proper parallel syntaxin-1A / synaptobrevin-2 subconfiguration within the ternary SNARE complex (*Figure 2*). Our results thus provide for an explanation of the dN-SB method. This method is related to an earlier report using a slightly different synaptobrevin fragment, the Vc peptide (*Melia et al., 2002*). There is independent evidence that improperly assembled SNAREs (including antiparallel subconfigurations) may be a feature of SNAREs in other contexts as well: for vacuolar fusion, proof-reading by the HOPS complex is essential for maximal fusogenic trans SNARE complex formation (*Zick and Wickner, 2014*). In any case, we used the dN-SB method for all subsequent smFRET studies in this work in order to ensure properly assembled SNARE complex.

## Two conformations of complexin bound to SNARE complex

Previously it had been suggested that the accessory domain of complexin-1 blocks binding of the membrane-proximal C-terminal part of synaptobrevin-2, preventing full zippering of the SNARE complex (*Giraudo et al., 2006*, *2009*), although subsequent experiments suggested that complexin-1 rather bridges partially assembled SNARE complexes (*Kümmel et al., 2011*). Our smFRET experiments now suggest that the accessory domain is involved in *either* interaction depending on the particular conformation of complexin-1 (*Figure 3*). By monitoring the position of the accessory domain with respect to the ternary SNARE complex, we found two major populations in the smFRET efficiency histograms. We note that ensemble averaging in previous FRET experiments probably masked the existence of the two populations (*Krishnakumar et al., 2011*; *Cho et al., 2014*). Our smFRET experiments have now resolved two conformations of complexin-1 that both exist when bound to full-length ternary SNARE complex. We ascribe the low and high FRET efficiency state to the *trans* and *cis* conformations of complexin-1, respectively. The superclamp mutant of complexin-1 slightly increases the population of the high FRET efficiency state, while the no-clamp mutant greatly decreases the population of the high FRET efficiency state (*Figure 3C,D*), suggesting that the accessory domain is important for the *cis* conformation of complexin-1.

## The *cis* conformation of complexin-1

By positioning FRET label pairs at the C-terminal end of the SNARE complex (*Figure 2B*), we monitored the effect of the *cis* conformation of complexin-1. Remarkably, a substantial fraction of the ternary SNARE complex exhibited a conformational change upon addition of complexin-1 as indicated by the emergence of a low FRET efficiency state (*Figure 4B*). This complexin-1 induced conformational change of the ternary SNARE complex depends on interactions involving the N-terminal, accessory, and core domains of complexin-1 (*Figure 4I*), suggesting an intimate interaction between all of these domains of complexin-1 and the ternary SNARE complex. Thus, the low FRET efficiency state likely corresponds to the *cis* conformation of complexin-1 when bound to the ternary SNARE complex. In support of this notion, mutations of the accessory helix of complexin-1 also had an effect on the conformations of complexin-1 when bound to the ternary SNARE complex. When attaching FRET label pairs on complexin-1 and syntaxin-1A within the complexin-1 / ternary SNARE supercomplex, the superclamp mutant slightly increased the population of the high FRET efficiency state (i.e., an increase of the *cis* conformation) compared to wildtype complexin-1 (*Figure 3D*). This effect correlates with a slight increase in the population of the low FRET efficiency state by the superclamp mutant when the FRET label pairs were attached to the C-terminal end of the SNARE complex (*Figure 4F*). Similarly, the effects for the no-clamp mutant were also correlated compared to wildtype complexin-1: there is a decrease of the *cis* conformation of complexin-1 (*Figure 3D*) and a decrease in the population of the low FRET efficiency state of the SNARE complex (*Figure 4F*). Therefore, the *cis* conformation (*Figure 3A*) of bound complexin-1 likely induces the conformational change of the ternary SNARE complex at the membrane-proximal C-terminal end (*Figure 4B*).

As independent evidence for a conformational change at the C-terminal end of the SNARE complex, $^1$H-$^{15}$N TROSY-HSQC NMR spectra with $^{15}$N labeled ternary SNARE complex and complexin-1 revealed interactions between the complexin-1 N-terminus with all components (syntaxin-1A, SNAP-25A, and synaptobrevin-2) at the C-terminal end of the ternary SNARE complex (*Figure 5b* in *Xue et al., 2010*). The interaction between complexin-1 and ternary SNARE complex was also examined by EPR (*Lu et al., 2010*). Most of the spin-labeled residues of the accessory domain of complexin-1 exhibited spectral broadening when quaternary complex was formed with the SNARE complex, likely due to formation of an α-helical conformation of the accessory domain of complexin-1. However, the authors noted that the spectra were sharper than those for solvent exposed residues on the surface of an α-helix that is part of a globular protein. Therefore, the observed sharpening could be due to motion between several conformations that are sampled by the accessory domain, consistent with the range of conformations that are suggested by our smFRET efficiency histogram (*Figure 3B*).

Spin-labeling of the C-terminal half of the cytoplasmic domain of synaptobrevin-2 indicated little effect on the EPR line widths for most of the labels (*Lu et al., 2010*), suggesting that the conformational change at the C-terminal end of the SNARE complex does not involve a dissociation of a part

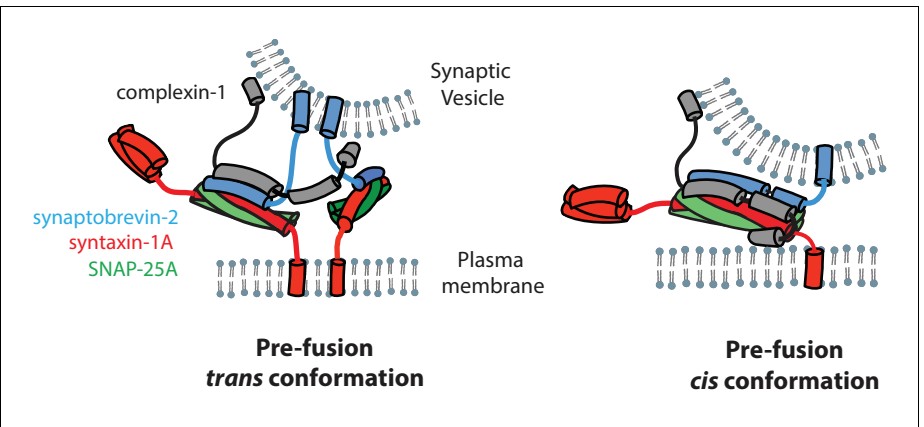

**Figure 7.** Models of the cis and trans conformations of complexin-1.

of synaptobrevin-2, but rather that synaptobrevin-2 is still interacting with the other components of the complexin-1 / ternary SNARE supercomplex.

The N-terminal domain (residues 1–27) of complexin-1 is important for activation of fast synchronous $Ca^{2+}$-triggered release and it interacts with the C-terminal end of the ternary SNARE complex (*Xue et al., 2007*, *2010*; *Maximov et al., 2009*). Since we have shown that the *cis* conformation of complexin-1 induces a conformational change at the C-terminal end of the SNARE complex through a mechanism that involves both the accessory and the N-terminal domains of complexin-1, we speculate that the *cis* conformation of complexin-1 is related to activation of $Ca^{2+}$-triggered release, although it could also play a role in regulation of spontaneous release (*Giraudo et al., 2006*; *2009*) (*Figure 7*). However, we note that mutations of the accessory domain of complexin-1 do not affect the activating function of complexin-1 compared to wildtype neurons in rescue experiments with complexin-1 knockdown (*Yang et al., 2010*, *2013*). Moreover, recent in vitro experiments revealed that the accessory domain is entirely dispensable for activation of $Ca^{2+}$-triggered synaptic vesicle fusion (*Lai et al., 2016*). These observations can be reconciled with the findings in this work by postulating that the interaction of the N-terminal domain of complexin-1 occurs for the *trans* SNARE complex that is juxtaposed between the membranes regardless of the accessory domain. In contrast when starting from fully assembled ternary SNARE complex, both the accessory and N-terminal domains are required to induce the observed conformational change at the C-terminal end of the SNARE complex. We speculate that the complexin-1 induced conformation of the SNARE complex may be related to the conformation of a *trans* SNARE complex that juxtaposes the synaptic vesicle and plasma membranes (*Figure 7*, right panel).

## The *trans* conformation of complexin

We studied interactions of complexin-1 in the *trans* conformation by first binding complexin-1 to a surface-tethered ternary SNARE complex via the central domain of complexin-1. The resulting complexin-1 / ternary SNARE supercomplex should allow the N-terminal, accessory, and C-terminal domains of bound complexin-1 to be accessible for other interactions when it is in the *trans* conformation. The binary (syntaxin-1A / SNAP-25A) SNARE complex bound weakly to the preassembled complexin-1 / ternary SNARE supercomplex via the accessory domain (*Figure 6E*). Together with previous results (*Kümmel et al., 2011*), one complexin-1 molecule in the *trans* conformation can bridge a variety of SNARE complexes. The correlations between complexin-1 mutants and this bridging interaction (*Figure 6C–E*) suggests that it is probably specific rather than merely a consequence of the flexible character of the binary SNARE complex that may expose hydrophobic elements in certain configurations (*Weninger et al., 2008*). We speculate that the *trans* conformation of complexin-1 is related to regulation of spontaneous fusion (*Krishnakumar et al., 2011*; *Kümmel et al., 2011*) (*Figure 7*, left panel). Our results also suggest that the two previously proposed SNARE complex 'clamping' models (*Giraudo et al., 2006*, *2009*; *Krishnakumar et al., 2011*; *Kümmel et al., 2011*) are not mutually exclusive (see both panels in *Figure 7*).

## The C-terminal end of the SNARE complex is more plastic than the N-terminal end

A conformational change at the membrane-proximal C-terminal end of the ternary SNARE complex has been independently observed upon αSNAP binding to ternary SNARE complex by smFRET experiments with labels attached to synaptobrevin-2 and SNAP-25A (*Ryu et al., 2015*). Similar to the complexin-1 induced conformational change at the C-terminal end of the ternary SNARE complex that we observe, a low FRET population was induced by αSNAP that increased as the αSNAP concentration was increased (compare Figure S4B in [*Ryu et al., 2015*] with *Figure 4E*). In contrast, no conformational change was observed at the N-terminal end of the SNARE complex upon αSNAP binding. Since two quite different factors (complexin-1 and αSNAP) can induce a conformational change at the C-terminal end of the SNARE complex, this suggests that the C-terminal end has different conformational properties than the N-terminal end. This finding is also consistent with multistage un-zipping of the SNARE complex by single molecule pulling experiments (*Gao et al., 2012*; *Marniemi et al., 1975*). Taken together, the observed plasticity of the membrane-proximal C--terminal end of the ternary SNARE complex likely has a functional role in membrane fusion, and the N-terminal domain of complexin-1 assists this process.

## Materials and methods

### Proteins: plasmids, expression, and purification

The cytoplasmic domain of rat syntaxin-1A (amino acid range 1–265), fused with a C-terminal biotinylation sequence (GLNDIFEAQKIEWHE), was cloned into the pTEV5 vector (*Rocco et al., 2008*) with a N-terminal TEV cleavable 6x-histidine tag. The constructs for rat SNAP-25A (amino acid range 1–206), for the cytoplasmic domain of rat synaptobrevin-2 (amino acid range 1–96), for the dN-SB fragment of synaptobrevin-2 (amino acid range 49–96), for rat complexin-1 with the N-terminal domain deleted (amino acid range 26–134), and for rat complexin-1 with both the N-terminal and accessory domains deleted (amino acid range 48–134) were also cloned into the pTEV5 vector that includes a N-terminal TEV cleavable 6 His-tag. Full-length complexin-1 (residues 1–134) was cloned into the pET28a vector (Novagen, EMD Chemicals, Gibbstown, NJ) that includes a N-terminal thrombin-cleavable 6x-histidine tag. Using site-directed mutagenesis, all cysteines of the wildtype proteins were mutated to serines. Unique labeling sites were introduced by cysteine mutations using QuikChange Kit (Agilent, Santa Clara, CA) at surface exposed positions based on the available crystal structures. We generated constructs for the following labeling sites, one at a time: syntaxin-1A S225C, E234C, S249C, S259C, synaptobrevin-2 S61C, A72C, A82C, K91C, and complexin-1 K26C, E24C. We also generated constructs for the superclamp mutant (D27L, E34F, R37A), no-clamp mutant (A30E, A31E, L41E, A44E), and 4M mutant (R48A, R59A, K69A, Y70A) of full-length complexin-1.

All proteins were expressed in *E. coli* BL21 (DE3) by growing the cells to $OD_{600}$ of about 0.8 at 37°C, then induced with 0.5 mM IPTG for 4 hr at 30°C. Syntaxin-1A biotinylation was performed in vivo by co-expression with a BirA gene engineered into pACYC184 (Avidity, Aurora, CO) and induced with 0.5 mM IPTG at 30°C in the presence of 0.1 mM biotin for 4 hr.

Cell pellets from 1 liter of culture were suspended in 40 ml of PBS (50 mM NaH2PO4, 300 mM NaCl, 0.5 mM DTT, pH 8.0) buffer and 0.5 mM PMSF supplemented with EDTA free Complete Protease Inhibitor Cocktail tablets (Roche, Basel, Switzerland). Cells were lysed by sonication at 1 s on and 2 s off pulse for 2 min at 50% power using Sonicator Ultrasonic Processor XL-2020 (Misonix, Farmingdale, New York) on ice water. Inclusion bodies were removed by centrifugation with JA-20 (Beckman, Coulter, Brea, CA) rotor at 20,000 rpm for 30 min. The supernatant was bound to Nickel-NTA agarose beads (Qiagen, Hilden, Germany) for 1 hr rotating at 4°C. The protein was washed extensively with PBS containing 20 mM imidazole while bound to Nickel-NTA agarose beads and eluted with the same buffer with 400 mM imidazole. 100 ug of TEV or thrombin protease was added and dialyzed overnight in 20 mM tris, 50 mM NaCl, 0.5 mM EDTA, 1 mM DTT, pH 8.0 to cleave off the N-terminal 6x-histidine tags. To separate the protease and the cleaved proteins, samples were purified using 1 ml HiTrap Q (GE Healthcare Bio-Sciences, Piscataway, NJ) with a linear gradient of 0.05 to 0.6 M NaCl in 20 mM tris, 0.5 mM TCEP, pH 7.5. The soluble synaptobrevin-2 fragment (amino acid range 1–96) does not bind to ion exchange columns. Therefore, cleaved samples were

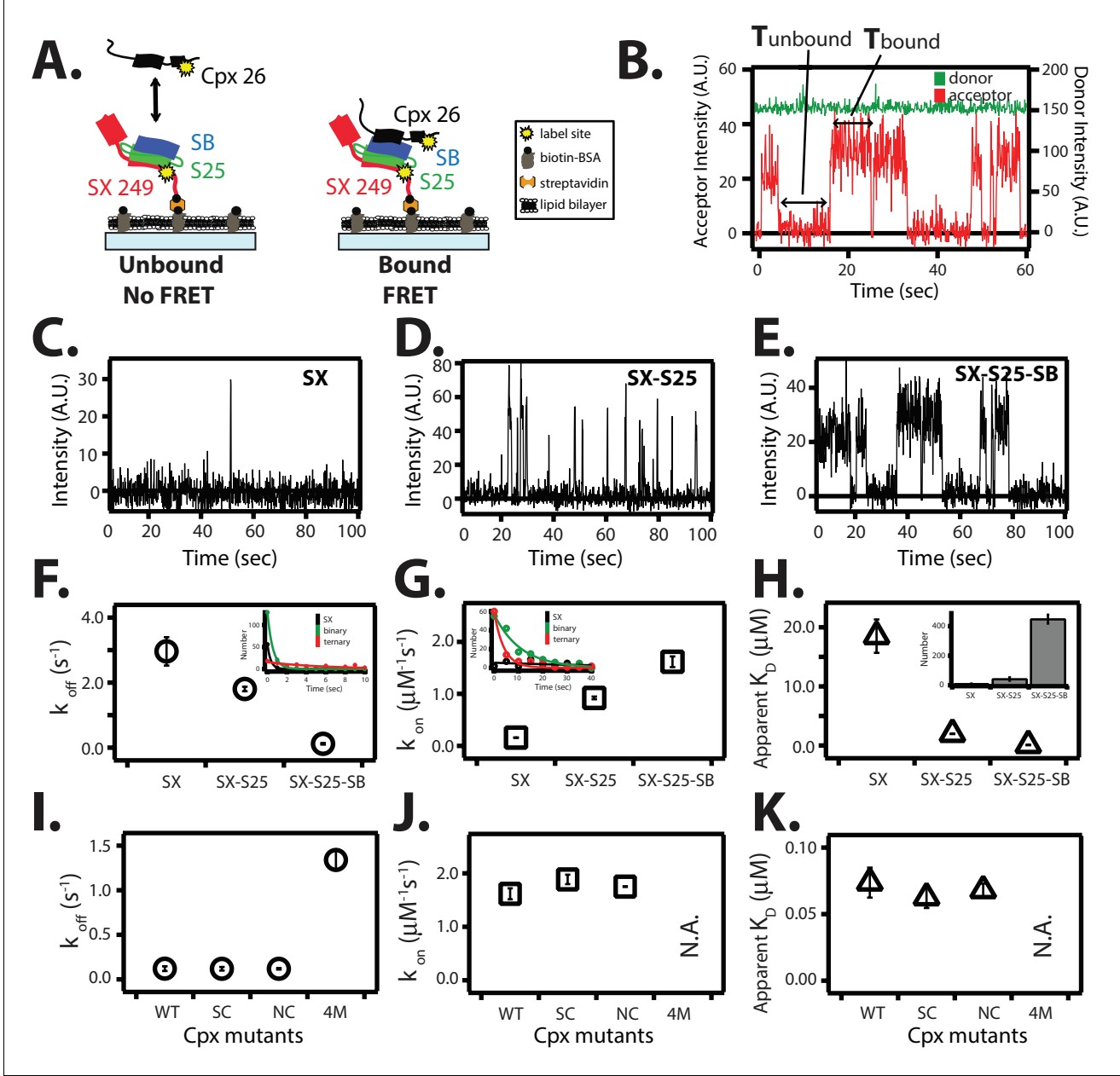

**Figure 8.** Single molecule detection of complexin-1 interacting with SNAREs. (**A**) Schematic of the smFRET binding assay. The cytoplasmic domain of syntaxin-1A (SX) was labeled with acceptor dye (Alexa 647) at residue position 249 (yellow dot) and surface-tethered through biotin-streptavidin (orange dot) linkage on a passivated microscope slide. Surface-tethered and labeled syntaxin-1A was used in isolation, or as part of a binary SNARE complex with SNAP-25A (SX-S25), or as part of a ternary SNARE complex with S25 and the cytoplasmic domain of synaptobrevin-2 (SB). Complexin-1 (Cpx) was labeled with donor dye (Alexa 555) labeled at residue position 26 (yellow dot) and added at a concentration of 100 nM in TBS buffer (20 mM Tris, 150 mM NaCl, pH 7.5) to the sample chamber. smFRET is expected if Cpx interacts with the surface-tethered SNARE complexes. A data summary table for all experiments in this figure is provided in *Figure 8—source data 1*. (**B**) Representative single molecule fluorescence intensity time traces showing individual binding events between ternary SNARE complex and Cpx. The donor dye fluorescence intensity (scale on the right y-axis) is colored green and the acceptor dye fluorescence intensity (scale on the left y-axis) is colored red. Due to the high concentration of the donor labeled proteins in solution, there is no significant effect on the donor intensity upon FRET with an acceptor dye. The stepwise increase in acceptor fluorescence intensity represents a bound state and gaps in between bound states represents unbound states. $T_{bound}$ and $T_{unbound}$ represent the dwell times of bound and unbound states, respectively. (**C–E**) Acceptor dye fluorescence intensities of Cpx molecules interacting with surface-tethered SX alone (**C**), binary SNARE (SX-S25) complex (**D**), and ternary SNARE complex (SX-S25-SB) (**E**). (**F,G**) Dissociation rates $k_{off}$ (open circles) and association rates $k_{on}$ (open squares) between Cpx and surface-tethered SX, binary SNARE complex (SX-S25), and ternary SNARE complex (SX-S25-SB). The insets show single exponential fits to histograms of the unbound and bound dwell times. Rates are calculated as described in Materials and methods. Error bars are

*Figure 8 continued on next page*

*Figure 8 continued*

standard deviations calculated from two subsets of an equal partition of the data (see data summary table in *Figure 8—source data 1*). (H) Apparent dissociation constant $K_D$ (= $k_{off}/k_{on}$) of Cpx binding to SX alone, binary SNARE complex (SX-S25), and ternary SNARE complex (SX-S25-SB). The inset shows bar graphs of the number of spots quantified from snapshots of the field of view of the acceptor channel corresponding to FRET events when complexin-1 interacts with the three different surface conditions. Error bars are standard deviations calculated from two subsets of an equal partition of the data (see data summary table in *Figure 8—source data 1*). (I,J) Dissociation rates $k_{off}$ (G, open circles) and association rates $k_{on}$ (H, open squares) between Cpx and its mutants (WT, wildtype; SC, superclamp; NC, no-clamp; 4M, mutation of the central complex domain that prevents SNARE complex binding, see Fig, 1A) and surface-tethered ternary SNARE complex. Rates are calculated as described in Materials and methods. Error bars are standard deviations calculated from two subsets of an equal partition of the data (see data summary table in *Figure 8—source data 1*). (K) Apparent dissociation constant $K_D$ (= $k_{off}/k_{on}$) for binding between ternary SNARE complex and complexin-1 or mutants. Error bars are standard deviations calculated from two subsets of an equal partition of the data (see data summary table in *Figure 8—source data 1*).

The following source data is available for figure 8:

**Source data 1.** Data summary table for the results shown *Figure 8F–K*.

purified using a Superdex 75 size exclusion column (GE Healthcare Bio-Sciences, Piscataway, NJ) in 20 mM tris, 150 mM NaCl, 0.5 mM TCEP, pH 7.5. Protein purity was checked using SDS-PAGE (>95%).

## Assembly and labeling of single SNARE complexes and complexin

Mutated proteins with single cysteine sites were labeled with Alexa 555 or 647 maleimide (Invitrogen, Carlsbad, CA) in 20 mM tris, 300 mM NaCl, pH 7.5 with 0.5 mM TCEP overnight at 4°C using a rotating platform. Free dye was removed by Sephadex G50 resin (GE Healthcare, Piscataway, NJ). The surface of the quartz microscope slide was coated with biotinylated BSA and passivated with 50 nm egg phosphatidylcholine liposomes (Avanti Polar Lipids, Alabaster, AL). Streptavidin was added to tether the biotinylated and Alexa 647 labeled cytoplasmic domain of syntaxin-1A to the surface, using conditions that produced a density of about 200–300 syntaxin-1A molecules per 45 x 90 $\mu m^2$ field of view. Binary SNARE complex (consisting of syntaxin-1A and SNAP-25) was formed by adding 1 $\mu$M of unlabeled SNAP-25 to the surface with tethered syntaxin-1A, incubated for 5 min, and then washing out the free SNAP-25A molecules that did not form complex. This method achieves the desired 1:1 stoichiometric ratio since the tethered syntaxin molecules are primary isolated at the low concentration (100–200 pM) and do not form homo-oligomeric species. To form ternary SNARE complex, 10 uM of the unlabeled cytoplasmic domain of synaptobrevin-2 (amino acid range 1–96) was added to the binary (syntaxin-1A / SNAP-25A) SNARE complex, incubated for 5 min, and then washed out to remove unbound synaptobrevin-2. For experiments with labeled synaptobrevin-2, samples were diluted to 1 nM to form ternary SNARE complex.

## The dN-SB method

An additional purification step in the presence of the denaturant urea was used previously in order to suppress improper SNARE subconfigurations that can occur when SNARE components are mixed (*Weninger et al., 2003*). However, the setup used in this work precludes the use of urea as an additional purification step since rinsing 7.5 M urea inside the microscope slide would disrupt the lipid bilayer surface along with the biotin-streptavidin linkage for protein tethering. Moreover, even with a denaturant purification step, there can be still up to 20 percent of improper subconfigurations (*Weninger et al., 2003*). Instead we included the 10 $\mu$M dN-SB fragment of synaptobrevin-2 (amino acid range 49–96) when ternary SNARE complex is assembled by adding synaptobrevin-2 to binary (syntaxin-1 / SNAP-25A) complex. After ternary SNARE complex formation, the unbound dN-SB fragments were removed by rinsing with TBS buffer (20 mM tris, 150 mM NaCl, pH 7.5). We refer to this method as the dN-SB method and used it in all single molecule experiments (*Figures 2–6*) in order to suppress improper SNARE subconfigurations.

## Single molecule fluorescence microscopy and data analysis

For the single molecule FRET experiments in *Figures 2–5*, we used protein free observation buffer (1% (w/v) glucose, 20 mM tris, 150 mM NaCl, pH 7.5) in the presence of oxygen scavenger (20 units/ml glucose oxidase, 1000 units/ml catalase) and triplet-state quencher (100 uM cyclooctatetraene).

For the single molecule real-time inter-molecular binding studies shown in *Figures 6* and *8*, 100 nM of Alexa 555 labeled complexin-1 was added to the observation buffer (1% (w/v) glucose, 20 mM tris, 150 mM NaCl, pH 7.5) in the presence of oxygen scavenger (20 units/ml glucose oxidase, 1000 units/ml catalase) and triplet-state quencher (100 uM cyclooctatetraene) in order to prevent fast photobleaching and blinking of the dye molecules.

Details of the single molecule fluorescence microscopy setup have been described elsewhere (*Choi et al., 2012*). Briefly, single molecule fluorescence intensities were recorded with a prism-type total internal reflection fluorescence (TIRF) microscope using 532 nm laser light (CrystaLaser, Reno, NV) excitation and detected by an Andor iXon EMCCD camera (Andor Technology, South Windsor, CT) at a frame rate of 10 Hz. The acceptor and donor fluorescence intensities were separated using a 640 nm single-edge dichroic beam-splitter (Shemrock, Rochester, NY) producing observation channels for both donor and acceptor fluorescence intensities. Fluorescence intensity movies were recorded for 1000 frames (100 s) until most of the dyes were photobleached with 532 nm laser light illumination. Data analysis was performed with the smCamera software from Taekjip Ha's group at University of Illinois and analyzed with scripts written for MATLAB (Mathworks).

For the smFRET experiments shown in *Figures 2–5*, fluorescence intensity histograms were generated by accumulating 50 frames from individual fluorescence intensity time traces (representative examples are shown in *Figure 4—figure supplement 1*) and converted to FRET efficiencies as described in the next section.

For the single molecule binding experiments shown in *Figures 6* and *8*, the binding events were characterized using hidden Markov modeling implemented in HaMMy software version 4.0 (*McKinney et al., 2006*) for a two-state system. High FRET represents the bound state and zero FRET represents the unbound state. The dwell times in the bound and unbound states were plotted in histograms and fitted to an exponential function to extract $k_{off}$ and $k_{on}$, respectively. $k_{on}$ is the rate of the unbound state divided by the concentration of the donor labeled protein in solution. The dissociation constant, $K_D$, is $k_{off}/k_{on}$.

## Calculation of FRET efficiency from observed donor and acceptor fluorescence intensities

The FRET efficiency (E) between two dye molecules with separation (r) is given by

$$E = \frac{1}{1 + \left(\frac{r}{R_0}\right)^6} , \qquad (1)$$

where the Förster radius $R_0$ is the distance when the efficiency of energy transfer is 50 percent. For the Alexa 555 and Alexa 647 dye pairs, $R_0$ is given as 5.1 nm as provided by the manufacturer of the dyes. Förster theory relates the fluorescence intensities of donor ($I_D$) and acceptor ($I_A$) dyes to the separation of the two dye molecules as (*Stryer and Haugland, 1967*; *Stryer, 1978*)

$$E = \frac{I_A}{I_A + I_D} . \qquad (2)$$

In the smFRET experiments shown in *Figures 2–4* (except *Figure 5*, see next section), FRET efficiency refers to

$$E = \frac{I_A - \beta(I_D - \alpha I_A)}{(I_A - \beta(I_D - \alpha I_A) + (I_D - \alpha I_A))} , \qquad (3)$$

where $\alpha I_A$ corrects for leakage of acceptor emission into donor channel and $\beta I_D$ corrects for leakage of donor emission into acceptor channel (*McCann et al., 2010*). The leakage of donor fluorescence into the acceptor channel was measured to be 1.7% and the leakage of acceptor fluorescence into the donor channel was 16.5%.

In *Tables 1* and *2* where we compare the FRET efficiencies to those estimated from known-crystal structures, we additionally applied a γ-correction, which accounts for detection efficiency and quantum yield of the two dye molecules (*Dahan et al., 1999*; *Ha et al., 1999*):

$$E = \frac{I_A - \beta(I_D - \alpha I_A)}{(I_A - \beta(I_D - \alpha I_A) + \gamma(I_D - \alpha I_A))},$$

(4)

where $\gamma = \Delta I_A / \Delta I_D$ is the γ-factor. We empirically estimated individual γ-factors from the changes in donor and acceptor fluorescence intensities before and after photobleaching of the acceptor dye molecule (*Ha et al., 1999*; *McCann et al., 2010*). The mean of the individual γ-factors was then used in *Equation (4)* to calculate the FRET efficiencies in *Tables 1* and *2*.

## Empirical γ-corrected smFRET efficiency histograms

We empirically determined the γ-factors for many individual fluorescence traces for the SFC1 and SFC2 label pairs in the presence and absence of 1 µM complexin-1 (*Figure 5C–D*). The mean γ-corrections are relatively small (*Table 2—source data 1*) for both dye pairs, arguing against a large influence on the sampling of orientations of the dye molecules by complexin-1 binding. Fluorescence anisotropy experiments (see below, and *Table 3*) also show that dye orientations are not affected by complexin-1 binding to the ternary SNARE complex. After application of the γ-corrections to the FRET efficiency histograms, the low FRET efficiency population is still observed (*Figure 5A–B*), suggesting that the induction of the low FRET efficiency population must be due to a genuine conformational change of the ternary SNARE complex that is induced by complexin-1. Moreover, for both SFC1 and SFC2 label pairs, the individual γ-factors for the low and high FRET populations are similar (*Figure 5C–D*).

## Estimation of FRET efficiencies from crystal structures

In *Tables 1* and *2* we estimated the distances for FRET pairs from the known high-resolution structures as previously described (*Choi et al., 2010*; *McCann et al., 2012*). Briefly, we used simulated-annealing molecular dynamics simulations starting from the crystal structures and conjugated fluorescent dye molecules to reduced cysteine residues (*Choi et al., 2010*). 100 simulations were performed where all protein atoms were fixed except for the fluorescent dye, which was allowed to freely rotate. From the distance measured between the mean dye position of Cy3 and Cy5, the FRET efficiency was calculated by using *Equation (1)*, where $R_0$ is given as 5.1 nm for the Alexa 555 and Alexa 647 dye pairs as provided by the manufacturer of the fluorescent dyes.

## Ensemble fluorescence anisotropy measurements

The steady-state anisotropy was measured relative with a Fluorolog spectrofluorometer (HORIBA scientific, Edison, NJ) relative to free Alexa 555 and 647 using an integration time of 2 s. For protein complexes singly labeled with Alexa 555, we used an excitation wavelength of 532 ± 5 nm and an emission wavelength of 575 ± 5 nm. For protein complexes that were singly labeled with Alexa 647, we used an excitation wavelength of 651 ± 5 nm and an emission wavelength of 671 ± 5 nm. The absorbance value for both dye labeled samples at the respective excitation wavelengths were adjusted to 0.05 AU for all samples to perform ensemble anisotropy measurements. SNARE proteins were individually purified and labeled with donor (Alexa 555) and acceptor (Alexa 647) dye molecule before complexes were formed. SNARE complex was assembled by mixing syntaxin-1 and SNAP-25 aliquots followed by addition synaptobrevin-2 fused to a N-terminal 6x-histidine tag at a ratio of 1:2:5, respectively, overnight at 4°C. The protein mixture was rebound to Nickel-NTA agarose beads (Qiagen, Hilden, Germany) and washed extensively with TBS (20 mM Tris, 150 mM NaCl, pH 7.5, 0.5 mM TCEP). Additional washing with TBS containing 7.5 M was used in order to favor the correct assembly and folding of the SNARE complex (*Weninger et al., 2003*). TEV was added to the eluted samples to cleave off the hexa-histidine tag of the synaptobrevin-2 construct for 1 hr at room temperature. The sample was then further purified by a Superdex 75 size exclusion column chromatography (GE Healthcare Bio-Sciences, Piscataway, NJ) in TBS in order to remove unbound SNAP-25A and synaptobrevin-2 molcules.

## Test of the surface tethering method

In order to test the surface tethering method used throughout this work (*Figures 2–6*), we compared complexin-1 binding with previous binding studies (*McMahon et al., 1995*; *Pabst et al., 2002*; *Bowen et al., 2005*; *Li et al., 2007*, *2011*). Syntaxin-1A was labeled with acceptor dye at the C--terminus at residue 249. The donor dye labeling site of complexin-1, residue 26, was chosen based on the crystal structure (PDB ID 1KIL) of the complexin-1 / ternary SNARE supercomplex (*Chen et al., 2002*) to produce high FRET efficiency when bound to the ternary SNARE complex (*Figure 8A,B*). Representative traces of real time measurements showed stochastic bursts of acceptor fluorescence upon complexin-1 binding to syntaxin-1A, binary, and ternary SNARE complexes (*Figure 8C–E*). We used hidden Markov modeling to extract dwell times of the bound and unbound states (*McKinney et al., 2006*). Histograms of individual dwell times were fitted with an exponential function to extract the kinetic rate constants $k_{off}$ and $k_{on}$ (*Figure 8F,G*). From the measured rate constants, we calculated equilibrium dissociation constants ($K_D$) (*Figure 8H*).

For the complexin-1 interaction with the ternary SNARE complex, the $k_{off}$ and $k_{on}$ rate constants were $0.12 \pm 0.03$ s$^{-1}$ and $1.62 \pm 0.1$ µM$^{-1}$s$^{-1}$, respectively, resulting in $K_D = 0.07 \pm 0.01$ µM (*Figure 8F–H*). These values agree reasonably well with previous single molecule measurements with ternary SNARE complex that used reconstituted full length syntaxin-1A (*Li et al., 2007*), validating the surface-tethering method used in this work. For the complexin-1 interaction to the binary SNARE complex the $k_{off}$ and $k_{on}$ rate constants were $1.8 \pm 0.07$ s$^{-1}$ and $0.92 \pm 0.03$ µM$^{-1}$s$^{-1}$, respectively, resulting in $K_D = 2.0 \pm 0.03$ µM, in good agreement with previous isothermal titration calorimetry (ITC) experiments ($K_D = 2.4 \pm 0.2$ µM, ref. [*Li et al., 2011*]). For the complexin-1 interaction to syntaxin-1A alone, the $k_{off}$ and $k_{on}$ rates constants were $2.96 \pm 0.44$ s$^{-1}$ and $0.16 \pm 0.001$ µM$^{-1}$s$^{-1}$, respectively, resulting in $K_D = 18.5 \pm 2.8$ µM. In summary, complexin-1 has a weak affinity to syntaxin-1A alone, and binds progressively more strongly from binary to ternary SNARE complex (*Figure 8H*).

We tested if the superclamp and no-clamp mutations affect binding to ternary SNARE complex in our single molecule binding assay (*Figure 8I–K*). As control we also tested the 4M mutant (*Figure 1A*). The kinetic rate constants of both the superclamp and no-clamp mutants were similar to that of wildtype complexin-1, resulting in $K_D = 0.063 \pm 0.01$ µM and $0.068 \pm 0.01$ µM, respectively (*Figure 8I–K*). As expected, the $k_{on}$ rate for the complexin-1 4M mutant was not measurable on the time scale of our experiment, and $k_{off}$ rate was $1.34 \pm 0.01$ s$^{-1}$, which is more than 10 times weaker than that for wildtype complexin-1. The lack of an effect of the no-clamp and superclamp mutants on the affinity between complexin-1 and SNARE complex suggests that under the particular conditions of this experiment, the binding is dominated by the interaction between the core domain of complexin-1 and ternary SNARE complex.

## Acknowledgements

We thank Jeremy Leitz for stimulating discussions and critical reading of the manuscript, Rabindra Shivnaraine for help with the steady state anisotropy measurements, Taekjip Ha for the smCamera program, and the National Institutes of Health for support (R37-MH63105 to ATB).

## Additional information

### Competing interests

ATB: Reviewing editor, *eLife*. The other authors declare that no competing interests exist.

### Funding

| Funder | Author |
| --- | --- |
| Howard Hughes Medical Institute | Axel T Brunger |
| National Institutes of Health | Axel T Brunger |

The funders had no role in study design, data collection and interpretation, or the decision to submit the work for publication.

## Author contributions
UBC, Conception and design, Acquisition of data, Analysis and interpretation of data, Drafting or revising the article; MZ, YZ, Analysis and interpretation of data, Drafting or revising the article; YL, Analysis and interpretation of data, Drafting or revising the article, Contributed unpublished essential data or reagents; ATB, Conception and design, Analysis and interpretation of data, Drafting or revising the article

## Author ORCIDs
Axel T Brunger, http://orcid.org/0000-0001-5121-2036

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
