## [Decision Letter]

Thank you for submitting your article "Complexin induces a conformational change at the C-terminal end of the SNARE complex" for consideration by *eLife*. Your article has been favorably evaluated by Randy Schekman (Senior editor) and three reviewers, one of whom is a member of our Board of Reviewing Editors.

The reviewers have discussed the reviews with one another and the Reviewing Editor has drafted this decision to help you prepare a revised submission.

Summary:

Complexin is thought to be a major regulator of neuroexocytosis and the structural basis for its regulation of SNARE-dependent membrane fusion is of great significance and interest. Choi et al. use single-molecule FRET to show that complexin can be in one of two conformations when bound to the SNARE complex. Importantly, the authors demonstrate that complexin has the capacity to disturb the C-terminal region of the SNARE complex, which potentially provides a structural basis for the "clamping mechanism". Further, the authors show that complexin bound to the ternary SNARE complex can make an intermolecular bridge to the syntaxin-SNAP25 binary complex, which supports the "trans-insertion model" proposed by Rothman and coworkers. Finally, the authors report a method using co-incubation with the C-terminal fragment of synaptobrevin to allow proper SNARE complex assembly of immobilized SNARE complexes.

The reported findings are very important for our current understanding of complexin function in regulating neurotransmission. Complexin clearly has multiple functions that have been localized to distinct domains of the protein, but contradictory models (all with substantial experimental support) have been put forth as the molecular mechanisms for these roles. The current study dramatically extends the molecular picture of the configurations of complexin interacting with the SNARE complex and the authors make convincing arguments about the connections between these configurations and complexin function. These results excitingly bridge several contradictory models, and thus the readers of *eLife* will receive the current work with broad interest.

However, the manuscript has a number of issues that should be resolved before further consideration.

Essential revisions:

Previous work by Lu et al. (JMB 2010; 396:602) is not cited in the manuscript, but has significant overlap with the results presented in this work. Intriguingly, their results and conclusions are contradictory to those presented in the current manuscript:

1) Lu et al. attached the nitroxides to positions 28, 35, 42 near or in the accessary helix region and investigated with EPR. In that work, Lu et al. did not observe two components that would reflect two different conformations of complexin when bound to the ternary SNARE complex. Although EPR is an ensemble technique, it is highly sensitive to protein conformational changes. Further, the method is fast enough to pick up the type of conformational changes discussed in the current manuscript.

2) In Lu et al.'s JMB paper they labeled the C-terminal positions of synaptobrevin 2. Again, complexin binding to the SNARE complex did not bring about spectral changes that reflects the large conformational change observed with FRET in this study.

Using single-molecule FRET data, the authors derive distances that are then compared to the distances obtained from the crystal structures. Distance information from smFRET experiments is notoriously unreliable; many factors such as local protein interactions and dye conformation play a role in the conversion between FRET values and distance. As is the convention in the field, it is much safer to use FRET values to classify conformational states and draw correlations between these values and different distances observed in crystal structures.

In discussing Figure 3, the authors make statements related to average distances corresponding to the two states in the smFRET experiments being equal to the distance derived from the ensemble-averaged experiments. This argument is mathematically incorrect: in a bulk-averaged FRET experiment one observes the average of two FRET values that each have a highly nonlinear dependence on the distance (1/R^6^), so that the average FRET value will certainly not be the same as a FRET value calculated using the average of the two distances. Furthermore, comparisons between FRET values derived from these two studies can only be made when the label sites and fluorophores/linkers are identical. This issue is connected to the inaccuracy of FRET experiments reporting on distance: one can only compare FRET values obtained with the exact same proteins and labeling, with conformation being the only difference.

The experiments described in Figure 5 are very insightful and present a nice way to probe the role of complexin in mediating interactions between two different SNARE complexes. In Figure 5, the authors show a raw trace of acceptor and donor intensity, but surprisingly FRET events correspond to only an increase in the acceptor intensity; the donor intensity remains unchanged. I assume that the 100 nM donor concentration in solution gives rise to a high background, but judging from the signal height and noise levels one should certainly see a drop in the donor signal when FRET occurs. The authors should explain this experiment in much more detail.

One concern with the experiments could be the use of fluorescent labels in the SNARE C-terminal region to probe conformational changes induced by complexin binding near the same region. Direct complexin contacts with the dyes or environmental changes due to close complexin binding could be speculated to impact FRET signals without conformational changes. The authors have taken substantial care to address such potential concerns. In particular, anisotropy studies of the fluorophores at these locations do not show changes with or without complexin (Figure 4—figure supplement 2), and γ factor analysis of FRET pairs Figure 4—figure supplements 3 and 4) also are unchanged with or without complexin. These results suggest dye rotational motion or quantum yields are not significantly altered by complexin binding. In addition, the authors might wish to further emphasize the observation that their measurement of binding kinetics and equilibrium binding constants (Figure 7) for labeled proteins that agree with literature values of unlabeled proteins suggests the fluorophores are not interfering with the biding interfaces. One more point is that the sum of donor and acceptor intensities for the traces shown in Figure 4—figure supplement 1 are all nearly the same in all of the FRET states. This suggests that there is no large Protein Induced Fuorescence Enhancement that is seen in some systems when a protein binds near a fluorophore (Hwang H. and Myong S. "Protein induced fluorescence enhancement (PIFE) for probing protein-nucleic acid interactions" Chem. Soc. Rev. 2014; 43:1221), further suggesting FRET changes are not an artifact.

The concentration dependence of complexin's effect on the C-terminal SNARE complex seems weaker (See Figure 4) than the equilibrium binding constant of complexin for the SNARE complex reported in Figure 7 and in the second paragraph of the subsection “Test of the surface tethering method by single molecule binding assay” as 70 nM. Am I misinterpreting the effect in Figure 4/E, or does this suggest that there is another process beyond simple bimolecular binding involved in this phenomenon of rearranging the SNARE c-terminal? Maybe this point could be emphasized in the paper if the authors agree.

In the subsection “Calculation of distances from FRET efficiency and error estimation”, is the width of a Gaussian FRET peak the best estimate for σ_sub_E? The width of FRET peaks for static conformations is typically dominated by statistical shot noise in the intensity signals. For the estimate of the error in the dye separation R, the uncertainty that is relevant is that in the accuracy of the peak value from the Gaussian fits used as the value E. Maybe this uncertainty in E is hard to determine as it is probably due to systematic variations and the width of the peak is some sort of estimate, but it is not clear how closely they are related. Maybe a cautionary comment is appropriate?

---

## [Author Response]

*Essential revisions:*

*Previous work by Lu et al. (JMB 2010; 396:602) is not cited in the manuscript, but has significant overlap with the results presented in this work.*

We apologize for the omission and have included this reference in the Discussion.

*Intriguingly, their results and conclusions are contradictory to those presented in the current manuscript:*

1) Lu et al. attached the nitroxides to positions 28, 35, 42 near or in the accessary helix region and investigated with EPR. In that work, Lu et al. did not observe two components that would reflect two different conformations of complexin when bound to the ternary SNARE complex. Although EPR is an ensemble technique, it is highly sensitive to protein conformational changes. Further, the method is fast enough to pick up the type of conformational changes discussed in the current manuscript.

Using electron paramagnetic resonance (EPR), Lu et al.(2010) monitored protein interactions between the accessory helix of complexin-1 and the C-terminal domain of ternary SNARE complex. By spin-labeling five residues of the accessory domain of complexin-1 (shown in their Figure 3), most of these spin labels showed spectral broadening when quaternary complex was formed with the SNARE complex. Lu et al.concluded that this broadening is due to formation of an α-helical conformation of the accessory domain of complexin-1. However, the authors noted that the spectra were sharper than those for solvent exposed residues on the surface of an α-helix that is part of a globular protein. Such sharpening could be due to motion between several conformations that are sampled by the accessory domain, consistent the range of conformations that are suggested by our smFRET efficiency histogram (Figure 3). Moreover, assuming that the *cis* conformation of complexin-1 would involve more tertiary interactions with the SNARE complex, it would result in a broader spectrum than for the *trans* conformation. Yet, the average of these spectra would be somewhere between these two cases and we believe that it would be difficult to uncover individual states of the system by these particular EPR experiments.

*2) In Lu et al's JMB paper they labeled the C-terminal positions of synaptobrevin 2. Again, complexin binding to the SNARE complex did not bring about spectral changes that reflects the large conformational change observed with FRET in this study.*

Lu et al.(2010) labeled five residues of synaptobrevin (A69C, Q76C, K83C, R86C, W89C). Only positions 69 and 76 showed EPR spectral broadening when complexin-1 bound to the SNARE complex (Figure 2 in Lu et al., 2010). _1_H-_15_N TROSY-HSQC NMR spectra of _15_N-labeled SNARE complex revealed chemical shifts for positions in all four SNARE components at the C-terminal end of the SNARE complex in the presence of full-length complexin-1 (Figure 5 in Xue et al., 2010), consistent with our observation of an interaction of full-length complexin-1 with the C-terminal end of the SNARE complex that depends on the presence of the N-terminal domain of complexin-1 (Figure 4). Thus, considering the NMR data by Xue et al., the absence of a change in line width observed by EPR is apparently consistent with the existence of interactions or conformational changes at the C-terminal end of the SNARE complex. However, we agree that the EPR results argue against dissociation of the C-terminal half of synaptobrevin-2 since the resulting motion of the C-terminal half of synaptovrevin-2 would be predicted to result in line sharpening. We apologize if we gave the impression of implying such a dissociation of the synaptobrevin-2 C-terminal half. Rather, our results suggest a conformational change of the SNARE complex where all the SNARE components are still interacting. We clarified the Discussion accordingly.

*Using single-molecule FRET data, the authors derive distances that are then compared to the distances obtained from the crystal structures. Distance information from smFRET experiments is notoriously unreliable; many factors such as local protein interactions and dye conformation play a role in the conversion between FRET values and distance. As is the convention in the field, it is much safer to use FRET values to classify conformational states and draw correlations between these values and different distances observed in crystal structures.*

We agree with the reviewers, and have removed all FRET-derived distances from the analysis of our data. Rather, we have restricted the presentation of our data in terms of FRET efficiencies. Comparisons to crystal structures were performed by estimating FRET efficiencies from the atomic coordinates of the structures as described in the Methods.

*In discussing Figure 3, the authors make statements related to average distances corresponding to the two states in the smFRET experiments being equal to the distance derived from the ensemble-averaged experiments. This argument is mathematically incorrect: in a bulk-averaged FRET experiment one observes the average of two FRET values that each have a highly nonlinear dependence on the distance (1/R^6), so that the average FRET value will certainly not be the same as a FRET value calculated using the average of the two distances. Furthermore, comparisons between FRET values derived from these two studies can only be made when the label sites and fluorophores/linkers are identical. This issue is connected to the inaccuracy of FRET experiments reporting on distance: one can only compare FRET values obtained with the exact same proteins and labeling, with conformation being the only difference.*

We agree with the reviewers. Consequently, we have also removed this particular comparison.

*The experiments described in Figure 5 are very insightful and present a nice way to probe the role of complexin in mediating interactions between two different SNARE complexes. In Figure 5, the authors show a raw trace of acceptor and donor intensity, but surprisingly FRET events correspond to only an increase in the acceptor intensity; the donor intensity remains unchanged. I assume that the 100 nM donor concentration in solution gives rise to a high background, but judging from the signal height and noise levels one should certainly see a drop in the donor signal when FRET occurs. The authors should explain this experiment in much more detail.*

We apologize for the confusion in Figure 3. The y-axis did not reflect the actual donor intensity scale. We have now plotted the actual donor intensity with respect to the scale that is provided on right y-axis of the panel (Figure 5). We note that due to the high concentration of the donor labeled proteins in solution, there is no significant effect on the donor intensity upon FRET with an acceptor dye.

*One concern with the experiments could be the use of fluorescent labels in the SNARE C-terminal region to probe conformational changes induced by complexin binding near the same region. Direct complexin contacts with the dyes or environmental changes due to close complexin binding could be speculated to impact FRET signals without conformational changes. The authors have taken substantial care to address such potential concerns. In particular, anisotropy studies of the fluorophores at these locations do not show changes with or without complexin (Figure 4—figure supplement 2), and γ factor analysis of FRET pairs Figure 4—figure supplements 3 and 4) also are unchanged with or without complexin. These results suggest dye rotational motion or quantum yields are not significantly altered by complexin binding. In addition, the authors might wish to further emphasize the observation that their measurement of binding kinetics and equilibrium binding constants (Figure 7) for labeled proteins that agree with literature values of unlabeled proteins suggests the fluorophores are not interfering with the biding interfaces. One more point is that the sum of donor and acceptor intensities for the traces shown in Figure 4—figure supplement 1 are all nearly the same in all of the FRET states. This suggests that there is no large Protein Induced Fuorescence Enhancement that is seen in some systems when a protein binds near a fluorophore (Hwang H. and Myong S. "Protein induced fluorescence enhancement (PIFE) for probing protein-nucleic acid interactions" Chem. Soc. Rev. 2014; 43:1221), further suggesting FRET changes are not an artifact.*

We thank the reviewers for the suggestions. We have emphasized the agreement between kinetics and binding values with literature values for unlabeled proteins, and we added the observation of the constant sum of donor and acceptor intensities.

*The concentration dependence of complexin's effect on the C-terminal SNARE complex seems weaker (See Figure 4) than the equilibrium binding constant of complexin for the SNARE complex reported in Figure 7 and in the second paragraph of the subsection “Test of the surface tethering method by single molecule binding assay” as 70 nM. Am I misinterpreting the effect in Figure 4/E, or does this suggest that there is another process beyond simple bimolecular binding involved in this phenomenon of rearranging the SNARE c-terminal? Maybe this point could be emphasized in the paper if the authors agree.*

We have conducted an additional experiment at 1 µM complexin-1 concentration which results in a conformational change of the SNARE complex that is statistically comparable to that observed at 10 µM (Figure 4), suggesting that complexin-1 binding reaches saturation at or before 1 µM. Thus, we do not believe there is sufficient evidence for another binding process.

In the subsection “Calculation of distances from FRET efficiency and error estimation”, is the width of a Gaussian FRET peak the best estimate for σ_sub_E? The width of FRET peaks for static conformations is typically dominated by statistical shot noise in the intensity signals. For the estimate of the error in the dye separation R, the uncertainty that is relevant is that in the accuracy of the peak value from the Gaussian fits used as the value E. Maybe this uncertainty in E is hard to determine as it is probably due to systematic variations and the width of the peak is some sort of estimate, but it is not clear how closely they are related. Maybe a cautionary comment is appropriate?

We apologize for the confusion. We used standard deviation of the Gaussian fit, σE=FWHM/(22ln2) which can be viewed as an estimate for the accuracy of the peak position. In any case, since we have removed the discussion of FRET-derived distances from this work, this analysis has also been removed.